



# The SDUST2022GRA global marine gravity anomalies recovered from radar and laser altimeter data: Contribution of ICESat-2 laser altimetry

Zhen Li[1], Jinyun Guo[1*], Chengcheng Zhu[2], Xin Liu[1], Cheinway Hwang[3], Sergey Lebedev[4], Xiaotao Chang[5], Anatoly Soloviev[4], Heping Sun[6]

[1] College of Geodesy and Geomatics, Shandong University of Science and Technology, Qingdao 266590, China
[2] School of Surveying and Geo-informatics, Shandong Jianzhu University, Jinan 250101, China
[3] Department of Civil Engineering, National Yang Ming Chiao Tung University, Hsinchu 300, Taiwan
[4] Geophysical Center, Schmidt Institute of Physics of the Earth, Russian Academy of Sciences, Moscow, Russia
[5] Land Satellite Remote Sensing Application Center, Ministry of Natural Resources, Beijing 100048, China
[6] State Key Laboratory of Geodesy and Earth's Dynamics, Innovation Academy of Precision Measurement Science and Technology, Chinese Academy of Sciences, Wuhan 430077, China

*Correspondence to*: Jinyun Guo (jinyunguo1@126.com)

**Abstract.** Global marine gravity anomaly models are predominantly recovered from along-track radar altimeter data. While remarkable advancements has been achieved in gravity anomaly modelling, the quality of gravity anomaly model remains constrained by the absence of across-track geoid gradients and the reduction of radar altimeter data, particularly in coastal and high-latitudes regions. ICESat-2 laser altimetry operates three-pair laser beams with a small footprint and near-polar orbit, enabling the determination of across-track geoid gradients and providing more valid observations in certain regions. The ICESat-2 altimeter data processing method is presented including the determination of across-track geoid gradients and the combination of along/across-track geoid gradients. A new global marine gravity model, SDUST2022GRA, is recovered from radar and laser altimeter data using different method for determining each altimeter data error. The accuracy and spatial resolution of SDUST2022GRA is assessed by published global gravity anomaly models (DTU17, V32.1, NSOAS22) and available shipborne gravity measurements. The accuracy of SDUST2022GRA is 4.43 mGal on a global scale, which is at least 0.22 mGal better than that of others models. Moreover, in local coastal and high-latitude regions, SDUST2022GRA achieves an accuracy improvement of 0.16-0.24 mGal compared to others models. The spatial resolution of SDUST2022GRA is approximately 20 km in a certain region, slightly better superior others models. These assessments suggests that SDUST2022GRA is a reliable global marine gravity anomaly model. By comparing SDUST2022GRA with incorporating ICESat-2 and SDUST2021GRA without ICESat-2, the percentage contribution of ICESat-2 to the improvement of gravity anomaly model accuracy is 13% in the global ocean region, and it is increasing with an proportion of ICESat-2 altimeter data in high-latitude and coastal regions. The SDUST2022GRA are freely available at the site of https://doi.org/10.5281/zenodo.8337387 (Li et al., 2023).



# 1 Introduction

Marine gravity is a critical piece of marine environmental information and accurately recovering marine gravity anomalies is essential for marine geophysics, marine geology, and marine dynamics (Hwang et al. 2014; Sandwell et al. 2014; Bidel et al.
2018; Wang et al. 2020). Since the late 1970s, satellite altimetry has been providing global sea surface height (SSH) observations associated with the time-invariant marine geoid. Because of the SSH characteristics of global coverage and consistent accuracy, the recovery of marine gravity anomaly model from satellite altimetry is an extremely important technique, as a supplement to in-situ gravity measurements (Andersen et al. 1998; Watts et al. 2020; Zhang et al. 2021).

The current gravity recovery method from altimetry is now quite stable. Normally, the gridded deflection of the vertical (DOV)
derived from along-track geoid gradients is used to recover marine gravity anomaly model by inverse Vening Meinesz formulae or Laplace's equation (Sandwell and Smith 1997; Hwang et al., 2002). Because of the accumulation of altimeter data and the improvements in altimeter data processing method, dozens of global marine gravity anomaly models have been published and continually refined  (Andersen et al., 2021; Zhu et al., 2020). Nonetheless, to investigate small-scale tectonics and mesoscale SSH variations, further improvements in the accuracy and spatial resolution of the marine gravity anomaly
model are still necessary (Yu et al., 2021; Sandwell et al., 2021).

The recovery of marine gravity anomalies is primarily from along-track radar altimeter data (Hwang et al. 2006; Andersen et al. 2010; Wu et al. 2019). Because of the satellite ground-tracks inclination of the north-south direction, the gridded DOV derived from along-track altimeter data has an unbalanced accuracy in the north-south component and the east-west component, with the accuracy of the north component being higher than the east component (Che et al., 2021; Jin et al. 2022). The
unbalanced accuracy in the components of the DOV severely restricts the improvement of the gravity anomaly model in accuracy and resolution (Hwang 1998, Annan and Wan 2021). The conception of twin-satellite altimetry and the simulated wide-swath altimeter data both aim to address the problem of unbalanced accuracy resulting from along-track altimeter data (Bao et al. 2013; Yu et al. 2021; Jin et al. 2022). Thus the addition of across-track altimeter data is necessary to refine the marine gravity anomaly model.

In addition, radar altimeter data is a crucial data source for the recovery of gravity anomalies, providing the centimeter-level accuracy of SSH observations (Vignudelli et al. 2011). In general, the conventional radar altimeter data has a large pulse-limited nadir footprint with a few kilometers in diameter (Escudier et al. 2018). The pulse-limited footprint, even for the SSH of Synthetic Aperture Radar (SAR) altimeter using Doppler shift technology, is a few hundred meters only in the along-track direction (Egido and Smith 2016; Vignudelli et al. 2019). Unfortunately, the radar echo signal used for SSH observations is
susceptible to interference from non-homogeneous reflective surfaces in coastal regions, leading to a degradation in SSH accuracy and a reduction in valid SSH observations (Hwang et al. 2006; Escudier et al. 2018). While altimeter data processing, such as waveform retracking, contributes to improving the quality of SSH, the accuracy of gravity anomalies recovered from degraded SSHs in coastal regions is still inferior to that in the open ocean (Passaro et al. 2018; Fernandes et al. 2021). Moreover,

only a few altimetry missions are capable of providing altimeter data in regions with latitudes larger than 66° due to orbital inclination design constraints (Li et al. 2022). The accuracy of the gravity anomaly model is also degraded in high-latitude regions (Andersen and Knudsen 2019; Ling et al. 2021). Therefore, the addition of altimeter data with new characteristics is crucial to improving the marine gravity anomaly model, particularly in coastal and high-latitude regions.

The ICESat-2 laser altimetry mission (Markus et al. 2017), launched successfully in September 2018, carries an advanced topographic laser altimeter system (ATLAS). The ATLAS provides three pairs of laser beams ground-track altimeter data, with approximately 3.3 km spacing for each pair in the across-track direction. The beam pair configuration of ICESat-2 allows for the determination of across-track height slope (Buzzanga et al. 2021). This provides an opportunity to mitigate the unbalanced accuracy of DOV caused by along-track altimeter data. In addition, the laser beam of ICESat-2 has a nominal 17 m diameter photon footprint, which makes SSH observations less susceptible to interference from non-homogeneous reflective surfaces compared to radar altimeter data. Although the small footprint is possible to cause adverse effects on the accuracy of SSH due to the surface ocean waves, it is especially useful for SSH observations in coastal regions (Wang et al. 2022; Wang and Sneeuw 2023). Furthermore, ICESat-2 provides near-global coverage of altimeter data with a 92° inclination, which is a valuable complement to radar altimeter data in high-latitude regions. Accordingly, SSH observations from ICESat-2 have been investigated in applications such as ocean topography recovery, DOV determination, and SSH anomalies variations examination, confirming that the quality is comparable to the best radar altimeter data (Yu et al. 2021; Che et al. 2021; Bagnardi et al. 2021). However, ICESat-2 altimeter data is rarely used in published global marine gravity anomaly models.

ICESat-2 laser altimeter data with unique characteristics compared to radar altimeter data motivates us to develop a new global marine gravity anomaly model and investigate its potential for gravity anomalies recovery. Firstly, the ICESat-2 altimeter data processing method is presented for the determination of across-track geoid gradients and the combination of along-track and across-track altimeter data. The new global marine gravity anomaly model (SDUST2022GRA) is recovered from multi-satellite altimeter data including ICESat-2 laser altimeter data. Secondly, the accuracy and spatial resolution of SDUST2022GRA is assessed by comparing published global marine gravity models (NSOAS22, DTU17, V32.1) and global available shipborne measurements. Finally, the contribution of ICESat-2 laser altimeter data is analysed by comparing SDUST2022GRA with previous version SDUS2021GRA without using ICESat-2 data.

## 2 Altimeter data and gravity anomalies data

### 2.1 ICESat-2 laser altimeter data

The ICESat-2 provides three pairs of laser beams, each pair consisting of a strong and a weak beam with about 4:1 energy ratio to measure Earth's surface elevation such as land/sea ice elevation, land/water vegetation elevation, and ocean elevation. For ocean elevation, ICESat-2 typically downlinks only strong beam data due to the lower surface reflectance. The ICESat-2



product, ocean elevation ATL 12 (level 3, version 5), provides along-track SSHs from three strong beams and is available via

NASA's Earth Science Data Systems (EarthDate, https://search.earthdata.nasa.gov/).

In ATL12, the SSH has been corrected for atmospheric delay, dynamic atmospheric errors, tidal errors, sea state bias, and other factors (Morison et al., 2021), where the ocean tide correction was derived from the global ocean tide model GOT4.8 with a resolution of 0.5° (Stammer et al., 2012). Because the recent global ocean tide model FES2014 (Carrere et al., 2015) has a resolution of 0.125° and is used for the L2P product of radar altimeter data, the correction from FES2014 instead of

GOT4.8 is used for the SSH from ICEsat-2, which is consistent with the product of radar altimeter data. In addition, due to the laser observation dependent on the weather conditions, the along-track ground distance of SSH observations is variable, between 70 m and 7 km. The SSH is referenced to the WGS84 reference ellipsoid (ITRF2014 reference frame, Morison et al., 2021). The ICESat-2 ground track of three strong beams from one cycle (91 days) is shown in Fig. 1.

### 2.2 Multi-satellite radar altimeter data

The multi-radar altimeter data used in SDUST2022GRA is similar to the previously published SDUST2021GRA (Zhu et al. 2022), which is primarily from altimetry missions after the 1990s. Although the ERS-1 altimeter data makes little contribution to the improvement of the gravity model, the geodetic mission (GM) altimeter data is used for the addition of data coverage, especially in high-latitude regions. In addition, the SAR altimeter data from new missions (Sentinel3A/3B, Sentinel-6A) is also used in SDUST2022GRA. The information about used altimeter data is presented in Table 1. The nominal tracks and

interleaved tracks from exact repeat missions (ERM) are labeled "_N" and "_I", respectively.

All SSHs of radar altimeter data were obtained from the non-time critical Level2+ (L2P, version 3) product which was the reprocessing Geophysical Data Records (GDR), except Sentinel-6A. The L2P is available at AVISO (https://www.aviso.altimetry.fr/). The Sentinel-6 SAR altimeter data is from the high-resolution non-time critical ocean surface

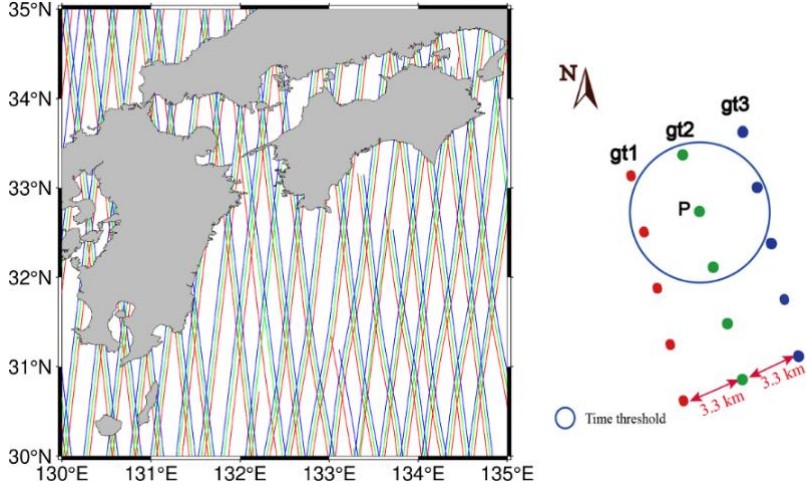

**Figure 1 ICESat-2 ground track of three strong beams (cycle_0011)**





topography product, which is available at NASA's EarthData (https://search.earthdata.nasa.gov/). All SSHs are from Ku-band altimeter data, except for the SSH of SARAL, which is from Ku-band altimeter data. The SSHs from radar altimeter data both are at 1 Hz sampling frequency and referenced to the WGS84 ellipsoid (CNES 2020).

**Table 1 Altimeter data information for global marine field recovery**

| Altimeter data | Observation Time (Cycles) | Orbit Inclination (°) | Repeat Period (d) | Ground track spacing in equator (km) |
|---|---|---|---|---|
| ICESat-2 | 2018.10-2022.04 (001-015) | 92 | 91 | 30/3/3 |
| SARAL/DP | 2016.07-2022.07 (100-162) | 98.55 | - | 5 |
| CryoSat-2/LRM | 2010.07-2020.06 (007-130) | 92 | 369 | 7.5±5 |
| HY-2A/GM | 2016.03-2020.06 (118-288) | 99.3 | 168 | 15 |
| Jason-2/GM | 2017.07-2019.10 (500-537/ 600-644) | 66 | 371/350 | 8.5/4 |
| Jason-1/GM | 2012.05-2013.06 (500-537) | 66 | 406 | 7.5 |
| ERS-1/GM | 1994.04-1995.09/1995.09- 1995.03 (030-040) | 98.52 | 168 | 8.3 |
| Sentinel-6A SAR | 2020.12-2022.07(004-062) | 66 | 10 | 293 |
| Sentinel-3A SAR | 2016.03-2022.08(001-088) | 98.64 | 27 | 104 |
| Sentinel-3B SAR | 2018.11-2022.07(017-067) | 98.64 | 27 | 104 |
| SARAL | 2013.03-2015.03(001-021) | 98.55 | 35 | 80 |
| HY-2A | 2014.04-2016.03(067-117) | 99.3 | 14 | 208 |
| HY-2B | 2019.12-2022.04(030-090) | 99.3 | | |
| Jason-3_N | 2016.02-2022.04(001-226) | | | |
| Jason-2_N | 2008.07-2016.10(001-303) | | | |
| Jason-2_I | 2016.10-2017.05(305-327) | | | |
| Jason-1_N | 2002.01-2009.01(001-259) | 66 | 10 | 316 |
| Jason-1_I | 2009.02-2012.03(262-374) | | | |
| T/P_N | 1992.09-2002.08(001-364) | | | |
| T/P_I | 2002.09-2005.09(369-479) | | | |
| Envisat_N | 2002.05-2010.10(006-093) | 98.55 | 35 | 80 |
| Envisat_I | 2010.11-2012.04(097-113) | | | |
| ERS-2 | 1995.05-2003.06(001-085) | 98.52 | 35 | 80 |
| GFO | 2001.01-2008.01(037-208) | 108 | 17 | 165 |



## 2.3 Global marine gravity anomaly models

The Earth Gravitational Field is typically used as the reference field in the recovery of gravity anomalies using the remove-restore technique. This technique is critical for the efficient computation of the short-wavelength gravity signal from SSH observations. The recently published XGM2019e is a combined global gravity model that combines the satellite gravity model GOCO06s, the marine gravity anomaly model DTU13, and gravity measurements over land and ocean (Zingerle et al. 2020). Gravity anomalies on 1′×1′ grid from XGM2019e up to degree and order 2190 are available via the International Centre for Global Earth Models (ICGEM, http://icgem.gfz-potsdam.de/calcgrid), which is used as the reference gravity field for the recovery of SDUST2022GRA.

The recently published global marine gravity anomaly models from altimetry were obtained for the assessment of SDUST2022GRA. The commonly recognized global marine gravity anomaly models are the Sandwell and Smith (S&S) series from the Scripps Institution of Oceanography (SIO) and the DTU series from the Technical University of Denmark. The version V32.1 for the S&S series (Sandwell et al. 2021) and DTU17 for the DTU series are publicly available. Other gravity models were also obtained including NSOAS22 (Zhang et al. 2022) recovered from incorporating HY-2 altimeter data and the SDUST2021GRA (Zhu et al. 2022) recovered by the improved data fusion method. These models are not yet using ICESat-2 laser altimeter data. The information on global marine gravity anomaly models is listed in Table 2. According to several studies, the root mean square of the difference between gravity anomaly models and shipborne gravity anomalies is approximately 3-5 mGal (Yu et al. 2022; Wan et al. 2022).

**Table 2 Global marine gravity anomaly models information**

| gravity anomaly models | Year | Reference gravity field | Coverage latitudes range | Main altimeter data |
|---|---|---|---|---|
| DTU17 | 2019 | EGM2008 | 90°S-90°N | Topex/Poseidon, Jason-1/2/3, ERS-1/2, Envisat, Cryosat-2 (LRM/SAR), SARAL/AltiKa |
| SIO V32.1 | 2022 | EGM2008 | 80°S-80°N | Topex/Poseidon, Jason-1/2/3, ERS-2, Envisat, Cryosat-2 (LRM/SAR), SARAL/AltiKa, Sentinel-3A/3B |
| NSOAS22 | 2022 | EGM2008 | 80°S-80°N | Geosat, ERS-1, Jason-1/2, Cryosat-2, SARAL/AltiKa, HY-2A/2B/2C/2D |
| SDUST2021GRA | 2022 | XGM2019e | 80°S-80°N | Topex/Poseidon, Jason-1/2/3, Envisat, Cryosat-2 (LRM), SARAL/AltiKa, HY-2A |



## 2.4 Shipborne gravity anomalies measurements

Shipborne gravity anomalies, as the in-situ gravity measurements, are also used to assess the accuracy and spatial resolution
of the gravity anomaly model recovered from altimetry. In general, shipborne gravity anomalies have a higher accuracy and
resolution than the gravity anomaly model on ship routes. Global shipborne gravity anomalies after the 1990s were obtained
from the U.S. National Centers for Environmental Information (NCEI), taking into account the effect of ship navigation for
the accuracy of gravity measurements. The data editing was performed to remove the gross error of shipborne gravity. Firstly,
we rejected those gravity measurement cruises with a large RMS and removed gravity measurement outliers larger than 3
times the standard deviation on each cruise by comparison with XGM2019e. Then, for each cruise gravity anomaly, a quadratic
polynomial was used to correct long wavelength system errors caused by the drift of the gravimeter, which is described in
Hwang and Parsons (1995). After data editing, the remaining shipborne gravity measurements are 7,012,812 points (486
cruises) with a rejection rate of 2.9%. The distribution of shipborne gravity anomalies is shown in Fig. 2.

Since global shipborne gravity anomalies are gathered from various agencies, the NCEI does not give information on the
precision of shipborne gravity measurements. The accuracy of shipborne gravity is verified by the discrepancies of gravity
anomalies at the crossover point. In the global ocean, the total number of crossover points is 49,277, and the RMS of
discrepancies is about 3.99 mGal. The accuracy of global shipborne gravity anomalies is about 2.82 mGal based on the error
propagation law, which is generally consistent with the shipborne gravimeter measurements of 1-3 mGal magnitude (Zaki et
al. 2022).

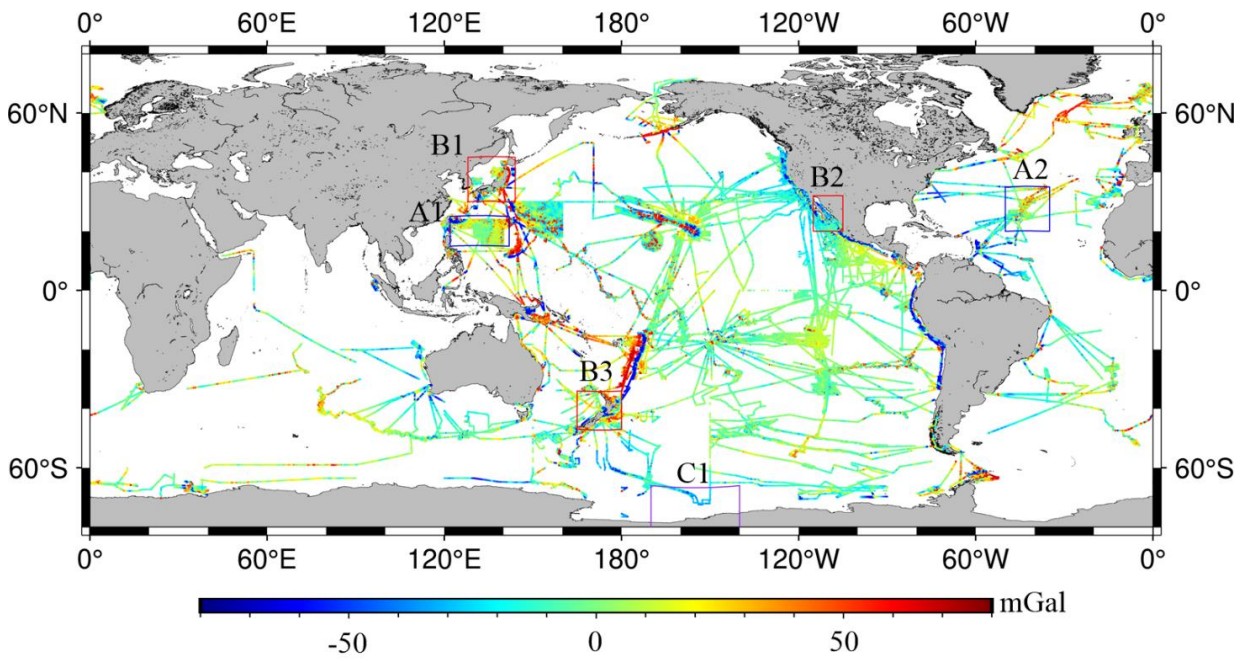


**Figure 2 Global available shipborne gravity anomalies from NCEI after the 1990s and local study regions**





In order to investigate the contribution of ICESat-2 to the recovery of local gravity anomalies, we selected six study regions where current or undersea features give rise to SSH variations. It includes two local open ocean regions (regions A1 and A2), three local coastal regions (regions B1, B2, and B3), and a local high-latitude region (region C1), as shown in Fig. 2. Regions

A1 and B1 are located in the Kuroshio Current region, and Region A2 is located in the North Atlantic with the Mid-Atlantic Ridge. Regions B2 and B3 are located in the Gulf of California and the New Zealand coastal regions, respectively. Region C1 is a local region of the Southern Ocean located in the eastern Ross Sea, accompanied by the Antarctic Circumpolar Current.

## 3 Marine gravity recovery methods

### 3.1 Multi-satellite radar altimeter data processing

The recovery of gravity anomalies from along-track radar altimeter data is a conventional method. Firstly, several errors are corrected for SSH observations, including instrument errors, atmosphere delay, and geophysical corrections. For ERM radar altimeter data, a simplified collinear adjustment is used to remove the residual time-variable error. This simplified method is carried out by averaging SSH observations from collinear altimeter data instead of data in the reference track, which is detailed in Rap et al. (1994) and Yuan et al. (2019). For GM radar altimeter data, Gaussian filtering is applied to remove the high-

frequency error.

Secondly, the residual geoid heights are determined by removing the mean dynamic topography model MDT_CNES_CLS18 (Mulet et al. 2021) and the reference geoid model from corrected SSHs. The residual along-track geoid gradient (GG) is derived by

$$e_{\alpha,res} = \frac{N_Q - N_P}{d_{PQ}} \tag{1}$$

where $e_{\alpha,res}$ is the residual GG with azimuth ($\alpha$) at the central location of Q and P, $N_Q$ and $N_P$ are residual geoid height at Q and P, respectively. $d_{PQ}$ is the spherical distance between two points.

The residual GGs can be converted to the gridded DOV by using the least squares collocation (LSC). The LSC is also a method of multi-satellite altimeter data fusion by determining the error variance from each altimeter data. The error variance of GG from each altimeter data can be derived using the error propagation law of Eq. (1) while ignoring the distance error of two

points, as

$$m_e^2 = \frac{m_{ssh,P}^2 + m_{ssh,Q}^2}{d_{PQ}^2} \tag{2}$$





where $m_e$ is the STD of GGs to determine the $C_{nn}$ in LSC, $m_{ssh,P}$ and $m_{ssh,Q}$ are the STD of SSH observations at P and Q, respectively.

The crossover discrepancies of SSH and the iterative method are applied to determine the GG errors from Ku-band and Ka-band altimeter data, respectively. For the crossover adjustment, a model of residual SSH errors is established using a combination function of a general polynomial and a trigonometric polynomial described in Huang et al. (2008), as

$$f(t) = a_0 + a_1(t - t_0) + \sum_{i=1}^{n}\left[C_i \cos(i\omega(t - t_0)) + S_i \sin(i\omega(t - t_0))\right] \tag{3}$$

where $f(t)$ is the SSH correction, $t$ is the observation time, $t_0$ and $t_1$ are the beginning and end observation times of each ground track, respectively. $\omega$ is the angular frequency ( $\omega = 2\pi/(t_1 - t_0)$ ), $a_0$, $a_1$, $C_i$, and $S_i$ are unknown parameters to be solved by the least squares method. The integer n is determined based on the number of crossover points.

The iterative method proposed by Zhu et al. (2020) is applied to determining the error of GG from the Ka-band altimeter data (SARAL/DP), which contributes to improving the accuracy of the marine gravity anomaly model. This method depends on the relationship among the error of altimeter-derived gravity, the error of GGs, and the average number of GGs, as:

$$D_{\Delta g} = \beta_0 + \beta_1 \frac{\rho}{m_e^2} \tag{4}$$

where $D_{\Delta g}$ is the error variance of altimeter-derived gravity, $\rho$ is the average number of GGs on $1' \times 1'$ grid, unknown parameters $\beta_0$ and $\beta_1$ can be solved by the least square method based on the error variance of altimeter-derived gravity, the error of GGs, and the average number from each Ku-band altimeter data.

The iterative equation for the error variance solution of GGs is

$$C_{nn}^{e,j+1} = \frac{\rho\beta_1}{D_{\Delta g,j} - \beta_0} \qquad\qquad j = 0,1,2\cdots \tag{5}$$

The initial value $D_{\Delta g,0}$ is determined using the gravity anomalies recovered from the initial error of GGs (SARAL/DP) derived by the RMS of crossover discrepancies. The termination condition of the iteration is that the difference between the adjacent error of GG ( $C_{nn}^{e,j+1}$ and $C_{nn}^{e,j}$ ) is less than a threshold.



## 3.2 ICESat-2 laser altimeter data processing

The ICESat-2 altimeter data processing method is presented for multiple beam observations. One major difference between the radar altimeter data and ICESat-2 laser altimeter data processing method is the determination of across-track GGs. In Eq. (1) of the last section, the along-track GG is determined from adjacent SSH observations of one track. But for the determination of across-track GG, it is necessary to select the associated SSHs from different beam tracks of ICESat-2. Otherwise, the GG derived from across-track altimeter data has an azimuth that deviates from the east-west direction, which cannot be used to mitigate the unbalanced accuracy of DOV.

Because three beams of ICESat-2 are not exactly simultaneous observations, the across-track GG is determined, according to the following steps. (1) One track with well observations from three beams is selected as the reference altimeter data. (2) Based on each observation time of the reference altimeter data, it needs to determine whether or not associated SSHs are available within a certain time threshold. (3) If available, the across-track GG is determined from associated SSHs with the closest time. A schematic diagram of determining across-track GGs from ICESat-2 altimeter data is shown in Fig. 3.

To determine across-track GGs, Gaussian filtering is likewise applied to along-track SSHs of ICESat-2. The LSC is used to fuse along-track GGs and across-track GGs based on the error variance of GGs. The error variance of across-track GGs is derived from the error of associated SSHs. Any two of three tracks from ICESat-2 can be used to determine the across-track GG, named gt12, gt23, and gt13, respectively.

## 3.3 Gravity anomalies recovery method

The used LSC for the determination of DOV is

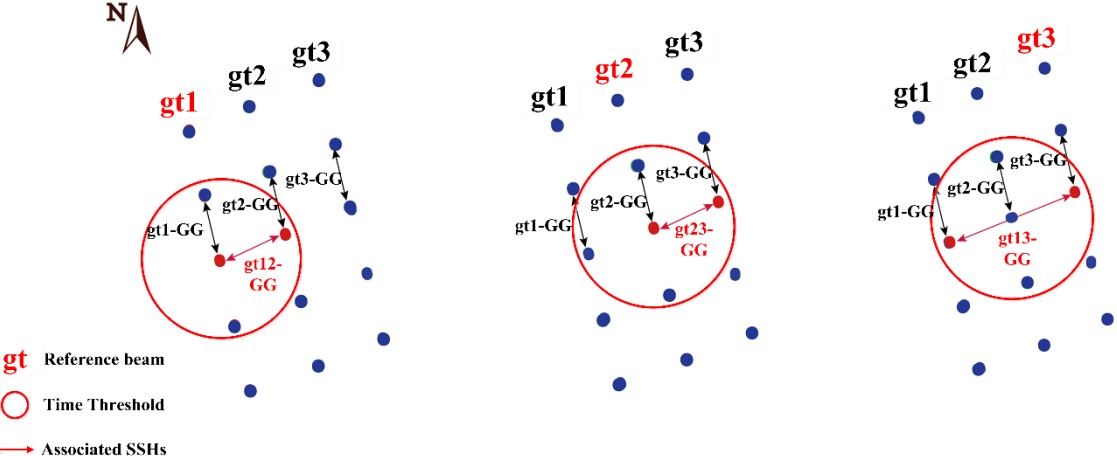

**Figure 3 The schematic diagram of determining the geoid gradients from ICESat-2**





$$\begin{pmatrix} \xi_{res} \\ \eta_{res} \end{pmatrix} = \begin{pmatrix} C_{\xi_e} \\ C_{\eta e} \end{pmatrix} \left( C_{ee} + C_{nn} \right)^{-1} e_{res,\alpha} \tag{6}$$

where $\xi$ and $\eta$ are the residual meridian and prime vertical component of the DOV, respectively. $C_{\xi_e}$ (or $C_{\eta e}$) is covariance

matrix for the meridian (or prime vertical) component of the DOV and GG, $C_{ee}$ is covariance matrix for the GG and GG. The

diagonal matrix $C_{nn}$ is the error variance of GGs. $e_{res,\alpha}$ is residual GGs.

The covariance function of residual disturbing potentials at the given distance can be calculated by errors of coefficients in the

potential set with Model 4 proposed by Tscherning and Rapp (1974). Because the longitudinal and transverse components are

isotropic, the covariance of longitudinal components $C_{ll}$ and transverse $C_{mm}$ for GGs can be derived by the covariance

function. Therefore, the covariance matrices ($C_{\xi_e}$, $C_{\eta e}$ and $C_{ee}$) are obtained by

$$\begin{cases} C_{\xi e} = -C_{ll} \cos \alpha_{PQ} \cos(\alpha_{eQ} - \alpha_{QP}) + C_{mm} \sin \alpha_{PQ} \sin(\alpha_{eQ} - \alpha_{QP}) \\ C_{\eta e} = -C_{ll} \sin \alpha_{PQ} \cos(\alpha_{eQ} - \alpha_{QP}) - C_{mm} \cos \alpha_{PQ} \sin(\alpha_{eQ} - \alpha_{QP}) \\ C_{ee} = C_{ll} \cos(\alpha_{eP} - \alpha_{PQ}) \cos(\alpha_{eQ} - \alpha_{PQ}) + C_{mm} \sin(\alpha_{eP} - \alpha_{PQ}) \sin(\alpha_{eQ} - \alpha_{PQ}) \end{cases} \tag{7}$$

where $\alpha_{eP}$ and $\alpha_{eQ}$ are azimuths of satellite ground track at the P and Q, respectively. $\alpha_{PQ}$ (or $\alpha_{QP}$) is the azimuth from P

to Q (Q to P).

The used inverse Vening-Meinesz formula for gravity anomalies grid model recovery is


$$\Delta g_p = \frac{\gamma_0}{4\pi} \iint_\sigma H^{'}(\psi) \left( \xi_q \cos \alpha_{QP} + \eta_q \sin a_{QP} \right) d_{\sigma_q} \tag{8}$$

$$H^{'}(\psi) = -\frac{\cos \frac{\psi}{2}}{2\sin^2 \frac{\psi}{2}} + \frac{\cos \frac{\psi}{2} \left( 2\sin \frac{\psi}{2} + 3 \right)}{2\sin \frac{\psi}{2} \left( \sin \frac{\psi}{2} + 1 \right)} \tag{9}$$

where $\gamma_0$ is the normal gravity. $H^{'}(\psi)$ is a kernel function of the spherical distance between two points, which is determined

by Hwang et al. (1998). $d_{\sigma_q}$ is the areal element of the unit sphere $\sigma$.

The gravity anomalies on the innermost-zone is derived by


$$\Delta g_{p,i} = \frac{s_0 \gamma_0}{2} (\xi_y + \eta_x) \tag{10}$$





$$s_0 = \sqrt{\frac{\Delta x \Delta y}{\pi}} \qquad (11)$$

where, $\xi_y$ and $\eta_x$ are obtained by numerical differentiations of the GGs. $s_0$ is the radius of the innermost zone. $\Delta x$ and $\Delta y$ are the grid intervals.

## 4 Gravity anomaly model recovery and assessment

### 4.1 Gravity anomalies recovered from ICESat-2

In the recovery of gravity anomalies from ICESat-2 altimeter data, SSHs at varying length scales from ICESat-2 are resampled to 1 Hz to integrate into radar altimeter data. The quality of SSHs and the accuracy of gravity anomalies recovered from SSHs at different sampling frequency are listed in Table 3. After resampling, the total number of SSHs is reduced, but the RMS of SSHs crossover discrepancies is improved by about 1 cm. Moreover, assessed by shipborne gravity and SIO V32.1, the RMS
of gravity anomalies from SSH at 1 Hz assessed by SIO V32.1 is slightly better than that of SSHs at varying length scales. Thus SSHs of ICESat-2 resampled at 1 Hz are used to recover global marine gravity anomalies.

The filtering radius is determined by the accuracy of gravity anomalies. For SSHs of ICESat-2, the maximum distance of along-track adjacent observations is about 7 km, so the filtering radius with a multiple of 7 km is applied to recover marine gravity anomalies from along-track altimeter data. When the filtering radius is 7 km, the RMS of difference between gravity
anomalies recovered from along-track altimeter data and shipborne gravity anomalies is 5.56 mGal, which is better than that without using Gaussian filtering (5.61 mGal) or a filtering radius of 14 km (5.58 mGal). Thus the filtering radius of 7 km is selected for the recovery of gravity anomalies from ICESat-2 along-track SSHs.

**Table 3 The quality of ICESat-2 SSHs and gravity models recovered from SSHs at varying length scales or at 1 Hz**

| SSHs at different sampling frequency | The number of SSHs | The RMS of SSH crossover discrepancies after adjustment (m) | The difference between Gravity anomalies recovered from ICESat-2 and Shipborne gravity (mGal) | | The difference between Gravity anomalies recovered from ICESat-2 and SIO V32.1 (mGal) | |
|---|---|---|---|---|---|---|
| | | | \|Max\| | RMS | \|Max\| | RMS |
| SSHs at varying length scales | 1457596 | 0.124 | 50.02 | 5.44 | 52.30 | 3.06 |
| SSHs at 1 Hz | 854533 | 0.115 | 49.54 | 5.42 | 52.01 | 2.89 |



The combination of along-track and different across-track GGs is discussed for the recovery of gravity anomalies, including gt1+gt2+gt3+gt12, gt1+gt2+gt3+gt23, gt1+gt2+gt3+gt13, and gt1+gt2+gt3+gt12+gt23. The difference between gravity anomalies recovered from ICESat-2 and shipborne gravity is listed in Table 4. The RMS of the gravity model recovered from gt1+gt2+gt3+gt13 is better than the result from gt1+gt2+gt3 by 0.14 mGal, which suggests that the accuracy of gravity model recovered from along-track GGs is improved by incorporating the gt13 across-track GGs. However, the accuracy of the gravity anomaly model recovered by incorporating the gt12 or gt23 across-track SSHs is not significantly increased. To analyze the reason, we compared the quality (total amount and standard deviation) of along-track and across-track GGs, as shown in Table 5. The total amount of gt13 GGs is generally consistent with gt12 and gt23, but the STD of difference between gt13 GGs and the reference gravity field is better than that of gt12 and gt23. This suggests that combining the lower-quality across-track altimeter data is likely to reduce the accuracy of gravity anomalies. Therefore, the combination of along-track and gt13 across-track GGs is used to recover marine gravity anomalies.

## 4.2 Global gravity anomalies recovered from all altimeter data

The error of GGs from each altimeter data is determined from SSH crossover discrepancies to fuse multi-satellite altimeter data, except for SDRAL/DP altimeter data. The crossover discrepancy is related to the time interval from two track observations. It is calculated from SSH observations within the smallest sub-cycle (approximately 30 days) of all altimetry missions, taking into account the number of crossover points and the effect of sea surface variation. For each ERM altimeter data, the crossover discrepancies are obtained from SSHs after collinear adjustment without the limit of time. The RMS of SSH crossover discrepancies is listed in Table 6.

The error of GGs from SARAL/DP altimeter data is determined by the iterative method. Unknown parameters ($\beta_0$ and $\beta_1$) in

**Table 4 Differences between ICESat-2 altimeter-derived gravity and ship-borne gravity (Unit: mGal)**

| Gravity anomaly model | Max | Min | Mean | STD | RMS |
|---|---|---|---|---|---|
| gt1+gt2+gt3 | 50.83 | -48.28 | -0.13 | 5.56 | 5.56 |
| gt12+gt1+gt2+gt3 | 49.35 | -48.18 | -0.10 | 5.66 | 5.62 |
| gt23+gt1+gt2+gt3 | 54.92 | -54.98 | -0.06 | 5.70 | 5.70 |
| gt12+gt23+gt1+gt2+gt3 | 47.07 | -46.75 | -0.07 | 5.65 | 5.65 |
| gt13+gt1+gt2+gt3 | 49.54 | -48.05 | -0.03 | 5.42 | 5.42 |

**Table 5 The number and STD of residual GGs from ICESat-2**

| Residual GGs | gt1 | gt2 | gt3 | gt12 | gt23 | gt13 |
|---|---|---|---|---|---|---|
| Number | 302407 | 250988 | 301138 | 202492 | 200312 | 209769 |
| STD($''$) | 1.93 | 1.88 | 1.91 | 2.66 | 2.75 | 1.94 |



280    iterative equation (Eq. 5) are solved by the error of gravity anomaly model, the error of GGs, and the average number within

$1'\times 1'$ grid from each Ku-band GM altimeter data, as shown in Table 7. The $\beta_0$ is 8.96 and the $\beta_1$ is -11.84 solved by the least

square method ( $R^2 = 0.98, RMS = 0.04$ ). The error of GGs determined by crossover discrepancies and the iterative method

is shown in Table 8. Based on the error of GGs (SARAL/DP) determined by the iterative method, the accuracy of gravity

anomalies recovered from SARAL/DP is improved by 9.1% compared to the result of the crossover discrepancies method.

285    Therefore, the GGs error variance of 2.37 is used for SARAL/DP altimeter data.

**Table 6 The RMS of SSH crossover discrepancies**

| Altimetry | Satellite Mission | Average along-track ground distance (km) | Crossover discrepancies (30 d) | |
|---|---|---|---|---|
| | | | RMS before adjustment (m) | RMS after adjustment (m) |
| Laser altimetry | ICESat-2/gt1 | 7.1 | 0.131 | 0.117 |
| | ICESat-2/gt2 | 7.1 | 0.128 | 0.109 |
| | ICESat-2/gt3 | 7.1 | 0.138 | 0.119 |
| GM (Radar altimetry) | SARAL/DP | 7.0 | 0.110 | 0.085 |
| | Cryosat-2 | 6.4 | 0.082 | 0.060 |
| | H2A | 6.5 | 0.103 | 0.076 |
| | J2 | 5.8 | 0.114 | 0.088 |
| | J1 | 5.8 | 0.108 | 0.079 |
| | E1 | 6.4 | 0.117 | 0.097 |
| ERM (Radar altimetry) | Sentinel-6A SAR | 5.8 | 0.022 | 0.013 |
| | Sentinel-3A SAR | 6.7 | 0.027 | 0.018 |
| | Sentinel-3B SAR | 6.7 | 0.035 | 0.026 |
| | SARAL | 7.0 | 0.034 | 0.020 |
| | HY-2A | 6.5 | 0.030 | 0.020 |
| | HY-2B | 6.5 | 0.032 | 0.024 |
| | T/P-Jason_A | 5.9 | 0.027 | 0.018 |
| | T/P-Jason_B | 5.9 | 0.026 | 0.019 |
| | Envisat_A | 7.5 | 0.033 | 0.022 |
| | Envisat_B | 7.5 | 0.042 | 0.024 |
| | ERS-2 | 6.6 | 0.040 | 0.034 |
| | GFO | 6.7 | 0.034 | 0.019 |



The accuracy and execution time of gravity anomalies recovery are impacted by the window length of LSC, which is connected to the amount of altimeter data. When the window length is 0.2°, the recovery of gravity anomalies is balanced between accuracy and execution time, as shown in Table 9. The global ocean region (0°E-360°E, 80°S-82°N) is divided into 144 (18×8, longitude by latitude) sub-regions (20°×20°) for the recovery of the global marine gravity anomaly model, and each sub-region is extended outward 1° (21°×21°) to mitigate boundary difference of gravity anomalies. The new global marine gravity anomaly model SDUST2022GRA (free air) on 1′×1′ grid is recovered from multi-satellite altimeter data, as shown in Fig. 4.

**Table 7 Altimeter gravity error, geoid heights error, and average number of geoid heights from Ku-band altimeter data**

| Gravity anomaly model | STD of difference between altimeter-gravity and shipborne gravity (mGal) | STD of difference between altimeter-gravity and SIO V32.1 (mGal) | Error variance of altimeter-gravity (mGal$^2$) | Error variance of GGs (mGal$^2$) | Geoid gradient average number |
|---|---|---|---|---|---|
| Jason-1/GM-derived | 5.59 | 3.09 | 9.00 | 7.84 | 0.146 |
| Jason-2/GM-derived | 5.53 | 3.11 | 8.70 | 9.86 | 0.229 |
| HY-2A/GM-derived | 5.42 | 2.97 | 7.67 | 5.81 | 0.465 |
| Cryosat-2-derived | 5.08 | 2.78 | 5.29 | 3.72 | 1.177 |

**Table 8 Marine gravity anomaly recovered from Ka-band altimeter data by different error of geoid gradients**

| Method | Error variance of GGs (mGal2) | STD of difference between altimeter-gravity and shipborne gravity (mGal) | STD from altimeter-gravity and SIO V32.1 (mGal) | STD of altimeter-gravity error (mGal) |
|---|---|---|---|---|
| The crossover discrepancies method | 6.35 | 5.19 | 2.77 | 2.42 |
| The iterative method | 2.37 | 5.00 | 2.75 | 2.20 |

**Table 9 The accuracy and execution time of gravity anomalies recovered by different window lengths in a sub-region (21°×21°)**

| Window length(°) | 0.1 | 0.2 | 0.3 | 0.4 |
|---|---|---|---|---|
| RMS(mGal) | 4.71 | 4.56 | 4.55 | 4.55 |
| Time(s) | 5530 | 141232 | 485218 | 1418156 |

Time was calculated based on CPU AMD Ryzen 5-3500X 6-Core @ 3.60GHz

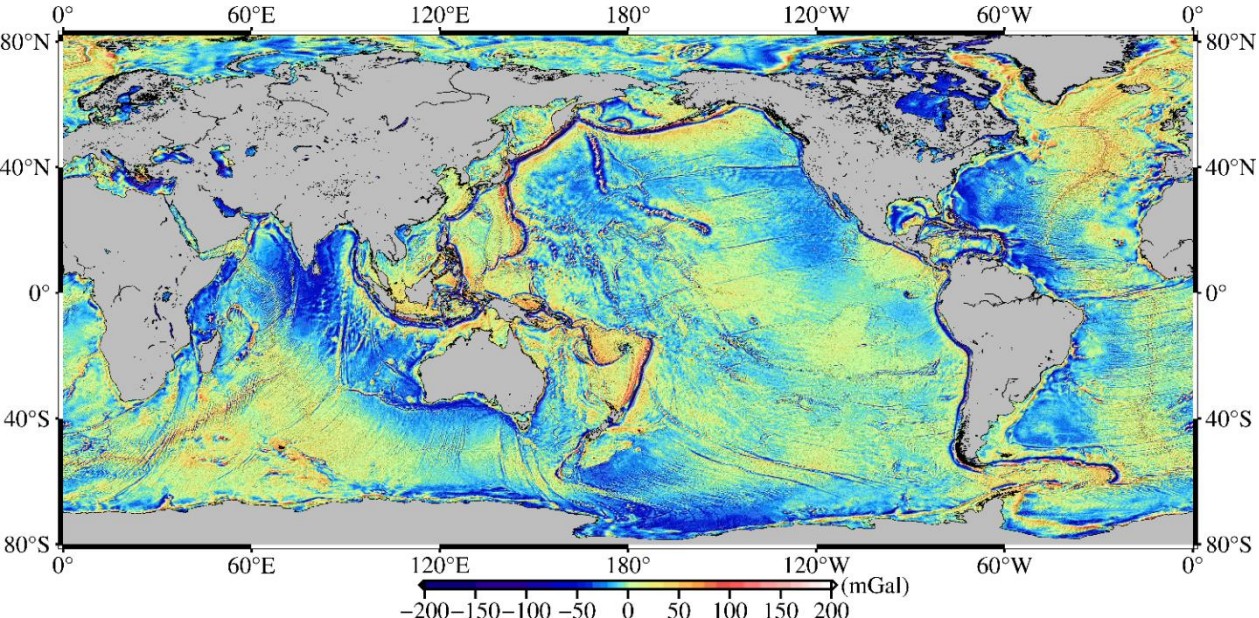

**Figure 4 The global marine gravity anomaly model SDUST2022GRA (free air) recovered from radar and laser altimeter data**

### 4.3 Assessment of gravity anomaly model accuracy

The accuracy of SDUST2022GRA is assessed by shipborne gravity anomalies in global and local ocean regions. The difference between global gravity anomaly models and global shipborne gravity anomalies is listed in Table 10. In four global gravity anomaly models, the accuracy of SDUST2022GRA and SIO V32.1 is generally better than that of NSOAS22 and DTU17, which primarily benefiting from the addition of new altimeter data. The accuracy of SDUST2022GRA in low-middle latitude regions is 4.43 mGal, which improved by 0.22 mGal in comparsion with the result of SIO V32.1. Moreover, the accuracy of all gravity anomaly models in low-middle latitude regions is significantly better than that in high-latitude regions. The main reason for the degraded accuracy of gravity model in high-latitude regions is the reduction of altimeter data (see Fig. 8).

The accuracy of gravity anomaly models is also analyzed in different local regions, including local open oceans (regions A1 and A2), local coastal regions (regions B1, B2, and B3), and local high-latitude region (region C1). The Mean and RMS of the difference between gravity anomaly models and shipborne gravity anomalies in local regions are presented in Table 11. Note that the shipborne gravity anomalies within 20 km from the coastline are used for the assessment of the gravity anomaly model in coastal regions.

The accuracy of all gravity models in open ocean regions is significantly better than that of gravity models in coastal regions and high-latitude regions. This shows that degraded SSH can significantly reduce the accuracy of gravity anomalies, especially in coastal regions and high-latitude regions. In local open ocean regions, SIO V32.1 and SDUST2022GRA have its own advantages resulting from unique improvement methods and the addition of altimeter data. For example, the SIO V32.1 benefit



from the improvement of along-track SSH gradients derived by two-pass waveform retracking, and the SDUST2022GRA benefits from the improvement of multi-altimeter data fusion and the addition of ICESat-2 across-track GGs. In local coastal

regions and high-latitude regions, the RMS of the SDUST2022GRA is 0.16-0.24 mGal better than that of SIO V32.1, which primarily benefiting from the valid observations from ICESat-2 laser beam. This assessment suggests that the SDUST2022GRA have a higher accuracy than others models in coastal regions and high-latitude regions. Thus SDUST2022GRA recovered by incorporating ICESat-2 laser altimeter data is a reliable global marine gravity anomaly model.

**Table 10 The difference between gravity anomaly models and global shipborne gravity (Unit: mGal)**

| Region | Model | Max | Min | Mean | STD | RMS |
|---|---|---|---|---|---|---|
| Global ocean [80°S, 82°N] | NSOAS22 | 99.46 | -81.17 | -0.10 | 5.73 | 5.73 |
| | DTU17 | 99.25 | -71.85 | -0.13 | 5.42 | 5.42 |
| | SIO V32.1 | 77.17 | -86.24 | -0.10 | 5.18 | 5.18 |
| | SDUST2022GRA | 96.79 | -68.51 | -0.08 | 5.07 | 5.07 |
| Low-middle latitude regions [60°S, 60°N] | NSOAS22 | 78.04 | -81.17 | -0.07 | 5.26 | 5.26 |
| | DTU17 | 78.44 | -71.85 | -0.12 | 4.89 | 4.89 |
| | SIO V32.1 | 76.25 | -86.23 | -0.06 | 4.65 | 4.65 |
| | SDUST2022GRA | 64.44 | -67.00 | -0.09 | 4.43 | 4.43 |
| High-latitude regions [80°S, 60°S)& (60°N, 82°N ] | NSOAS22 | 99.46 | -70.56 | -0.47 | 9.76 | 9.77 |
| | DTU17 | 99.25 | -68.48 | -0.25 | 9.82 | 9.82 |
| | SIO V32.1 | 77.17 | -76.54 | -0.51 | 9.53 | 9.54 |
| | SDUST2022GRA | 96.79 | -68.48 | -0.26 | 9.69 | 9.69 |

**Table 11 The Mean and RMS of difference between gravity anomaly models and shipborne gravity in local regions (Unit: mGal)**

| Local region | | NSOAS22 | | DTU17 | | SIO V32.1 | | SDUST2022GRA | |
|---|---|---|---|---|---|---|---|---|---|
| | | Mean | RMS | Mean | RMS | Mean | RMS | Mean | RMS |
| Region A1 | Open ocean | 0.15 | 3.58 | 0.10 | 3.24 | -0.10 | 3.15 | 0.20 | 3.04 |
| Region A2 | | -0.41 | 5.13 | -0.41 | 4.29 | 0.14 | 3.78 | 0.01 | 4.01 |
| Region B1 | Coastal region | -1.51 | 8.47 | -1.81 | 7.21 | 0.10 | 6.25 | -0.16 | 6.08 |
| Region B2 | | -0.86 | 10.66 | -1.41 | 10.33 | -0.56 | 7.85 | -0.57 | 7.69 |
| Region B3 | | 0.10 | 12.12 | -1.24 | 11.25 | -0.67 | 10.32 | -0.68 | 10.10 |
| Region C1 | High-latitude region | 0.33 | 5.86 | 0.15 | 5.36 | 0.12 | 5.38 | 0.12 | 5.14 |



## 4.4 Assessment of gravity anomaly model resolution

The spatial resolution of the gravity anomaly model in the local region is generally determined by spectral coherence analysis along the track of shipborne gravity measurements (Marks et al. 2016). The wavelength corresponding to the coherence magnitude squared (CMS) value of 0.5 is considered to be the highest spatial resolution of a gravity anomaly model. Three
cruises shipborne gravity anomalies are used to determine the spatial resolution of SDUST2022GRA, SIO V32.1, and DTU17, as shown in Fig. 5. The CMS between gravity anomaly models and shipborne gravity is shown in Fig. 6.

The wavelength derived from SDUST2022GRA with a CMS value of 0.5 is 18.6 km, 20.7 km, and 20.4 in a local open ocean region, high latitude region, and coastal region, respectively. The spatial resolution of SDUST2022GRA in the open ocean is generally superior to that in high latitude and coastal regions, which is largely related to the density of the altimeter data. The
average number of altimeter data within the 1′×1′ grid in the open ocean is significantly higher than that in high latitude and coastal regions (see Fig. 8). The spatial resolution of the SDUST2022GRA is approximately 20 km in a certain region, which is slightly better than DTU17 and SIO V32.1. Although SDUST2022GRA is recovered by incorporating ICESat-2 altimeter data, the resolution is not significantly increased compared to DTU17 and SIO V32.1. Therefore, it is still a challenge to achieve a gravity anomaly model with a spatial resolution of a few kilometers from current altimeter data, and anticipation for
the future wide-swath altimeter data from the SWOT altimetry mission (launch on 16/12/2022).

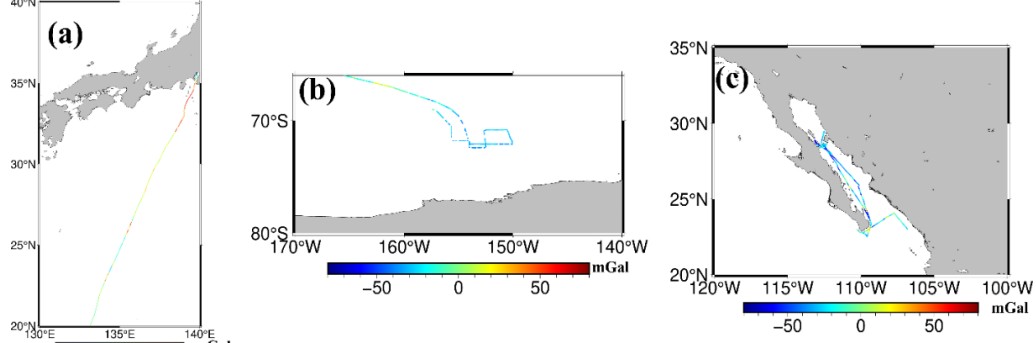

**Figure 5 Shipborne gravity (used to determine CMS) of different cruises. a: the jare33l1 with average distance interval of 0.45 km. b:. the ew9201 with average distance interval of 0.80 km. c: the moce05mv with average distance interval of 0.22 km.**

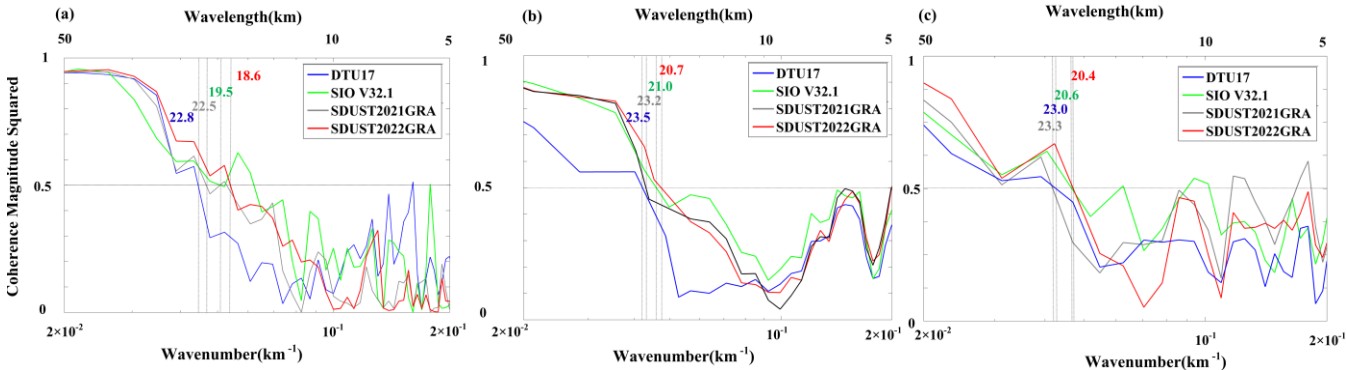



**Figure 6 The CMS between the gravity model and shipborne gravity of different cruises: (a) jare33l1, (b) ew9201, (c) moce05mv.**

## 5 Assessment of ICESat-2 contribution

### 5.1 Contribution on model accuracy

The importance of ICESat-2 to the recovery of the gravity anomaly model is investigated by contrasting ICESat-2 with each radar altimeter data. It is widely recognized that the GM radar altimeter data play an important role in the recovery of marine

gravity anomalies. The role of ICESat-2 in the ranking of GM altimeter data is also determined according to the gravity anomaly model recovered by removing each GM altimeter data from all altimeter data in the local region (120°E -140°E, 20°N-40°N). The RMS of the difference between each gravity anomaly model and shipborne gravity anomalies is listed in Table 12. It is evident that the SARAL/DP and Cryosat-2 altimeter data provide a major improvement in the accuracy of the gravity anomaly model. The contribution of ICESat-2 altimeter data to the improvement outperforms that of other GM

altimeter data. This suggests that the ICESat-2 altimeter data is comparable to most GM altimeter data and is an extremely important dataset to improve the marine gravity anomaly model. In addition, all ERM data is also crucial to enhance the global marine gravity anomaly model.

The contribution of ICESat-2 to the improvement in the accuracy of gravity anomalies is assessed by comparing SDUST2022GRA recovered by incorporating ICESat-2 and SDUST2021GRA without ICESat-2. Although the ICESat-2

altimeter data is not used in DTU17 or SIO V32.1, the gravity recovery method of DTU17 or SIO V32.1 is different from SDUST2022GRA. The difference between SDUST2022GRA and SIO V32.1 (or DTU17) also includes the variation caused by different methods. Because the SAR altimeter data from S3A/3B and S6A with sparse coverage is used in SDUST2022GRA, the improvement in the accuracy of gravity anomaly model is initially determined. The RMS of the difference between the gravity anomaly model recovered by only incorporating SAR altimeter data and shipborne gravity anomalies is 4.64 mGal,

which is consistent with that without SAR altimeter data (SDUST2021GRA). This suggests that SAR altimeter data makes little contribution to the improvement of the gravity anomaly model. The difference between SDUST2022GRA and SDUST2021GRA can be attributed primarily to the addition of ICESat-2 altimeter data.

The percentage contribution of ICESat-2 to the improvement of the gravity anomaly model is defined as
$\dfrac{RMS_{\text{SDUST2022GRA}} - RMS_{\text{SDUST2021GRA}}}{RMS_{\text{SDUST2022GRA}} - RMS_{\text{reference\_field}}} \times 100\%$ , representing the ratio of the improvement of the gravity model recovered

by incorporating ICESat-2 to the improvement of the gravity model recovered from all altimeter data, as shown in Table 13. The percentage contribution of ICESat-2 is achieved at about 13% in global ocean regions while the number of SSH from ICESat-2 takes up 10% of all radar altimeter data. And the percentage contribution is high in low-middle latitude regions and high-latitude regions, even though there is a small improvement in high-latitude regions. This indicates that the ICESat-2





altimeter data has a high contribution to the improvement of the gravity anomaly model recovered from current radar altimeter
data.

Moreover, the percentage contribution of ICESat-2 is also determined for different local regions, including the open ocean, coastal regions, and the high latitude region. In those regions, the difference between the SDUST2022GRA and the SDUST2021GRA is shown in Fig. 7. The RMS of the difference between both models is 0.83 mGal and 0.72 mGal in local open ocean regions A1 and A2, respectively. In coastal regions, note that the RMS is only derived from the difference within
20 km of the coastline. It is 1.29 mGal, 0.98 mGal, and 1.26 mGal in local coastal regions B1, B2, and B3, respectively. And the RMS is 1.22 mGal in the local high latitude region C1. These results indicate that the variation in the accuracy of the gravity model is visible by incorporating ICESat-2 altimeter data, especially in coastal and high-latitude regions.

The percentage contribution of ICESat-2 in local coastal and high-latitude regions is generally higher than that in open ocean regions, as shown in Table 14. In order to investigate the reason for the variation in the percentage contribution of ICESat-2
in different regions, the average number of GGs from all altimeter data within the $1'\times1'$ grid is calculated, as presented in Fig. 8. Because the average number from all radar altimeter data is relatively low in high latitude and coastal regions, it is increased by 50% and 58% from the addition of ICESat-2, respectively. In open ocean regions, however, it is only increased by 21%, which lower than that in high latitude and coastal regions. This suggests that the high percentage contribution of ICESat-2 to the improvement is correlated with the increased proportion of the average number. In addition, 42% and 35% of the ICESat-
2 altimeter data are located in $1'\times1'$ grid where no radar altimeter data is available in high latitude and coastal regions, but only

**Table 12 Ranking of altimeter data contribution to gravity anomaly model recovery**

| Removed altimeter data | SARAL/ DP | Cryosat-2 | ICESat-2 | All ERM | HY-2A/ GM | Jason-2/ GM | Jason-1/ GM | ERS-1/ GM | No |
|---|---|---|---|---|---|---|---|---|---|
| RMS (mGal) | 4.70 | 4.66 | 4.64 | 4.64 | 4.61 | 4.60 | 4.59 | 4.57 | 4.57 |
| RMS difference (mGal) | 0.13 | 0.09 | 0.07 | 0.07 | 0.04 | 0.03 | 0.02 | 0 | - |

**Table 13 The percentage contribution of ICESat-2 altimeter data in global ocean region**

| Region | RMS of Reference Earth gravitational model (mGal) | RMS of SDUST2021GRA (mGal) | RMS of SDUST2022GRA (mGal) | $\triangle RMS_{ICE/SAR}$ (mGal) | $\triangle RMS_T$ (mGal) | Percentage Contribution |
|---|---|---|---|---|---|---|
| Global ocean | 5.97 | 5.19 | 5.07 | 0.12 | 0.90 | 13% |
| Low-middle latitude regions | 5.52 | 4.63 | 4.43 | 0.20 | 1.09 | 18% |
| High-latitude regions | 9.95 | 9.73 | 9.69 | 0.04 | 0.26 | 15% |





9% of the ICESat-2 is located in 1′×1′ grid in open ocean region. This suggests that the ICESat-2 altimeter data presents the complementary characteristics of SSH coverage due to the reduction of radar altimeter data in high latitude and coastal regions.

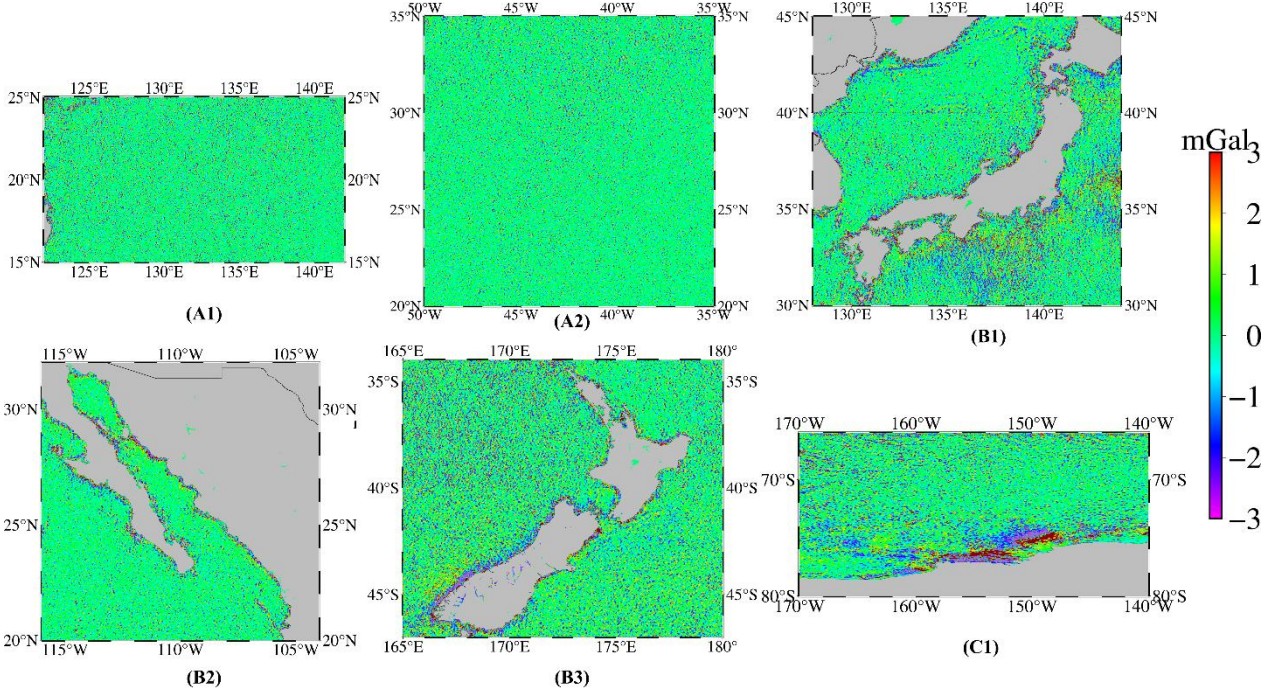

**Figure 7 The difference between SDUST2022GRA and SDUST2021GRA in different local regions.**

**Table 14 The percentage contribution of ICESat-2 altimeter data in different local regions**

| Local region | RMS of Reference Earth gravitational model (mGal) | RMS of SDUST2021GRA (mGal) | RMS of SDUST2022GRA (mGal) | $\triangle RMS_{ICE/SAR}$ (mGal) | $\triangle RMS_T$ (mGal) | Percentage contribution |
|---|---|---|---|---|---|---|
| Region A1 | 3.68 | 3.12 | 3.04 | 0.08 | 0.64 | 12% |
| Region B | 4.76 | 4.07 | 4.01 | 0.06 | 0.75 | 8% |
| Region C | 8.25 | 6.40 | 6.08 | 0.32 | 2.17 | 15% |
| Region D | 10.30 | 7.98 | 7.69 | 0.28 | 2.61 | 11% |
| Region E | 11.31 | 10.51 | 10.10 | 0.41 | 1.21 | 34% |
| Region F | 6.26 | 5.32 | 5.14 | 0.18 | 1.12 | 15% |

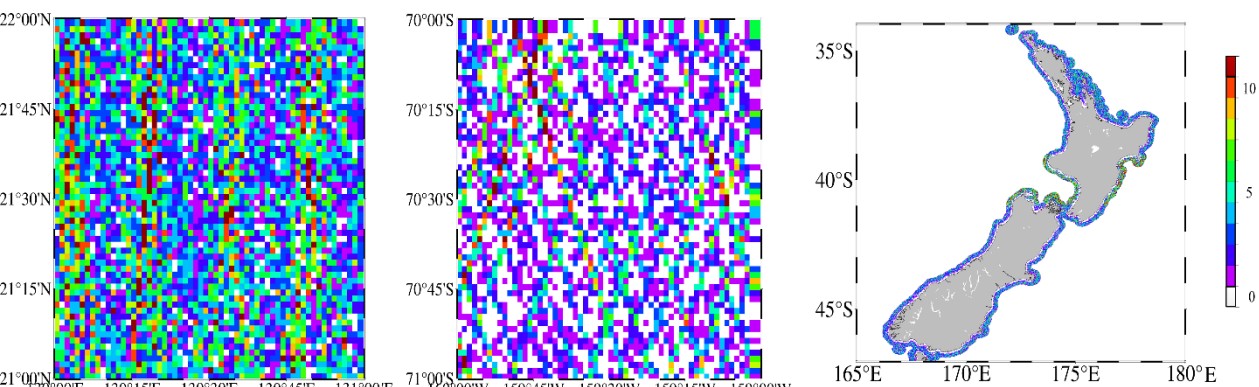

Figure 8 The number of SSHs within the 1′×1′ grid in different local regions. a: open ocean with average number of 3.5. b: the high latitude region with average number of 2.1. c: coastal region with average number of 1.9.

## 5.2 Contribution on model resolution

To analyze the contribution of ICESat-2 to the spatial resolution of gravity anomaly model, we also compared the spatial resolution of SDUST2022GRA and the SDUST2021GRA, as shown in Fig. 6. The wavelength corresponding to the CMS of 0.5 is reduced from 22.5 km to 18.6 km in the local open ocean, from 23.2 km to 20.7 km in the local high-latitude region, and from 23.3 km to 20.4 km in the local coastal regions, respectively. The spatial resolution of the gravity anomaly model is slightly increased by incorporating the ICESat-2 in a certain local region. However, the increased signal of gravity anomaly is mainly from the power on wavelength bands greater than 18 km. This suggests that the SSHs of ICESat-2 can improve the marine gravity anomaly model at wavelength >18 km, but the contribution to higher resolution should be small.

## 6 Data availability

The global marine gravity anomaly model, SDUST2022GRA, is available at the ZENODO repository, https://doi.org/10.5281/zenodo.8337387 (Li et al., 2023). The dataset includes global marine free-air gravity anomalies (WGS84 ellipsoid) in NetCDF file fortmat (i.e., vector of latitudes, vector of longitudes, and matrix of gravity anomalies).

## 7 Conclusions

The recovery of the global marine gravity anomaly model is mainly from along-track radar altimeter data. The advanced ICESat-2 laser altimetry mission provides SSHs from multiple beams and valid observations in high latitude and coastal regions. This provides the potential to mitigate unbalanced accuracy caused by along-track altimeter data and increase altimeter data in high latitude and coastal regions. The gravity anomalies recovery method from across-track altimeter data is proposed and used to recover gravity anomalies from ICESat-2. The new global marine gravity model SDUST2022GRA is recovered

from multi-satellite radar altimeter data and ICESat-2 laser altimeter data. According to SDUST2022GRA and previously
published SDUST2021GRA without ICESat-2, we investigate the contribution of ICESat-2 to the recovery of the global
marine gravity anomaly model, including the combination of along-track and across-track altimeter data, as well as the addition
of SSHs in high latitude and coastal regions.

The accuracy and spatial resolution of SDUST2022GRA are assessed by global shipborne gravity anomalies and published
global marine gravity anomaly models (DTU17 and SIO V32.1). The accuracy of SDUST2022GRA is 4.43 mGal in low-
middle regions, which improved by 0.22 mGal at least in comparsion with others published gravity anomaly models. Moreover,
the accuracy of SDUST2022GRA is 0.16-0.24 mGal better than that of others models in the local coastal and high-latitude
regions. Based on the spectral coherence analysis, SDUST2022GRA has a spatial resolution of about 20 km in a certain region,
which is slightly better than the resolution of DTU17 and SIO V32.1. These results show that SDUST2022GRA is a reliable
global marine gravity anomaly model.

The recovery of gravity anomalies only from ICESat-2 demonstrates that incorporating across-track altimeter data can improve
the accuracy of gravity anomalies from along-track altimeter data as envisaged. The combination of along-track and across-
track altimeter data from ICESat-2 plays an important role in the recovery of gravity anomalies and can be considered an
important dataset following the SARAL/DP and Cryosat-2 altimeter data. By comparing SDUST2022GRA and previous
version SDUST2021GRA without ICESat-2, the percentage contribution of ICESat-2 to the improvement of gravity anomaly
model accuracy is 13% in the global ocean region, and it is high with an increasing proportion of altimeter data in high-latitude
and coastal regions. In addition, the ICESat-2 altimeter data is effective in improving the spatial resolution of gravity anomaly
model greater than 20 km, which is similar to the best radar altimeter data.

**Author contribution.** All authors contributed to recover global marine gravity anomaly model and editing the manuscript.

**Competing interests.** All authors have no competing interests to declare that are relevant to the content of this article.

**Acknowledgments.** We are very grateful to NASA's Earth Science Data Systems and AVISO for providing altimeter data,
and NCEI for providing global shipborne gravity measurements.

**Financial support.** This work was partially supported by the National Natural Science Foundation of China (grants 42192535,
42274006, 42242015)

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
