# Peer review of "The SDUST2022GRA global marine gravity anomalies recovered from radar and laser altimeter data: Contribution of ICESat-2 laser altimetry"

_Earth System Science Data, 2023_

## Referee Comment (RC2)

**The SDUST2022GRA global marine gravity anomalies recovered from radar and laser altimeter data: Contribution of ICESat-2 laser altimetry**

Zhen Li[1], Jinyun Guo[1*], Chengcheng Zhu[2], Xin Liu[1], Cheinway Hwang[3], Sergey Lebedev[4], Xiaotao Chang[5], Anatoly Soloviev[4], Heping Sun[6]

[1] College of Geodesy and Geomatics, Shandong University of Science and Technology, Qingdao 266590, China

[2] School of Surveying and Geo-informatics, Shandong Jianzhu University, Jinan 250101, China

[3] Department of Civil Engineering, National Yang Ming Chiao Tung University, Hsinchu 300, Taiwan

[4] Geophysical Center, Schmidt Institute of Physics of the Earth, Russian Academy of Sciences, Moscow, Russia

[5] Land Satellite Remote Sensing Application Center, Ministry of Natural Resources, Beijing 100048, China

[6] State Key Laboratory of Geodesy and Earth's Dynamics, Innovation Academy of Precision Measurement Science and Technology, Chinese Academy of Sciences, Wuhan 430077, China

*Correspondence to*: Jinyun Guo (jinyunguo1@126.com)

In this manuscript, 'across-track' should be corrected to 'cross-track'.

**Abstract.** Global marine gravity anomaly models are predominantly recovered from along-track radar altimeter data. While remarkable advancements has been achieved in gravity anomaly modelling, the quality of gravity anomaly model remains constrained by the absence of across-track geoid gradients and the reduction of radar altimeter data, particularly in coastal and high-latitudes regions. ICESat-2 laser altimetry operates three-pair laser beams with a small footprint and near-polar orbit, enabling the determination of across-track geoid gradients and providing more valid observations in certain regions. The ICESat-2 altimeter data processing method is presented including the determination of across-track geoid gradients and the combination of along/across-track geoid gradients. A new global marine gravity model, SDUST2022GRA, is recovered from radar and laser altimeter data using different method for determining each altimeter data error. The accuracy and spatial resolution of SDUST2022GRA is assessed by published global gravity anomaly models (DTU17, V32.1, NSOAS22) and available shipborne gravity measurements. The accuracy of SDUST2022GRA is 4.43 mGal on a global scale, which is at least 0.22 mGal better than that of others models. Moreover, in local coastal and high-latitude regions, SDUST2022GRA achieves an accuracy improvement of 0.16-0.24 mGal compared to others models. The spatial resolution of SDUST2022GRA is approximately 20 km in a certain region, slightly better superior others models. These assessments suggests that SDUST2022GRA is a reliable global marine gravity anomaly model. By comparing SDUST2022GRA with incorporating ICESat-2 and SDUST2021GRA without ICESat-2, the percentage contribution of ICESat-2 to the improvement of gravity anomaly model accuracy is 13% in the global ocean region, and it is increasing with an proportion of ICESat-2 altimeter data in high-latitude and coastal regions. The SDUST2022GRA are freely available at the site of https://doi.org/10.5281/zenodo.8337387 (Li et al., 2023).

The 13% improvement in the marine gravity field is significantly impressive, but ICESat-2 only improved by 0.12 mgal, which could mislead readers.

[Figure]

The authors confused the 'DOV' and 'geoid gradient (GG)' concepts. In this manuscript, DOV is geoid gradients, but they are opposite. The along-track GGs (and cross-track GG from ICESat-2) are used to determine the north and east components of GG on a regular grid using LSC. Therefore, this sentence should be: 'Normally, the north and east components 
[revised manuscript text omitted]

575

---

## Author Comment (AC1)

**Comment:** This paper presents significant new material by introducing the SDUST2022GRA, a new 1 arcmin global marine gravity anomaly model, along with various comparisons to state-of-the-art global gravity field models and in situ observations. The authors leverage radar and laser altimetry to harness the strengths of individual missions enabled by advanced altimeter technologies. The beam pair configuration of ICESat-2, which allows for the determination of cross-track height slopes, offers an opportunity to enhance the precision and spatial resolution of the gravity anomaly model. This article provides valuable insights into gravity anomaly recovery using along-track and cross-track data. Overall, the paper introduces significant advancements in the field, but addressing the following concerns would enhance its clarity and impact.

Reply: Thanks very much for your valuable suggestions and comments. These comments play an important role in revising the paper and improving the quality of the paper. We have revised our manuscript according to your comments. Below, we describe in detail the changes to the manuscript on a point-by-point basis.

Concerns:

1.  Section 4.1 emphasizes the critical role of altimeter data downsampling in mitigating high-frequency noise, which is essential for ensuring data accuracy and reliability. For radar altimeter data, a 1 Hz sampling rate is typically employed. The authors note that ICESat-2 laser altimeter data exhibit varying length scales, ranging from 70 m to 7 km. However, the downsampling method applied to the ICESat-2 laser altimeter data is not specified. Clarification on this method is essential, as it significantly impacts the quality of the SDUST2022GAR model.

Reply: Thanks for your suggestion. Resampling is an important method in altimeter data processing. We have added the introduction about resampled method in Section 3.2 and presented the precision of SSHs before and after resampling in Section 4.1.

Section 3.2 ICESat-2 laser altimeter data processing

The ICESat-2 SSH observations at varying length scales is resampled at 1 Hz for each beam to achieve a uniform distribution of SSHs. In the resampling, SSHs at varying length scales are fitted using a quadratic polynomial in latitude to reduce the effect of the high-frequency noise and outlier. Each 1-s SSHs is used to solve polynomial coefficients and then produced SSHs in the median of the latitude. When the number of observations is less than the minimum number for the solution of polynomial coefficients, the 1-s SSHs are averaged directly to 1 Hz. The used quadratic polynomial function of latitude is (Yu and Hwang, 202)

$$l_i + v_i = a\varphi_i^2 + b\varphi_i + c \qquad (6)$$

where $l_i$ is the SSH observation at point $i$ with in a time threshold, $v_i$ is the residual at point $i$, $\varphi_i$ is the latitude at point $i$, and $a$, $b$, $c$ are the coefficients of the quadratic polynomial.

Section 4.1 Gravity anomalies recovered from ICESat-2

For the recovery of gravity anomalies from ICESat-2 altimeter data, SSHs at varying length scales from ICESat-2 are resampled to 1 Hz to integrate into radar altimeter data. The quality of SSHs and the accuracy of gravity anomalies recovered from SSHs at different sampling frequency are listed in Table 3. After resampling, the total number of SSHs is reduced, but the RMS of SSHs crossover discrepancies

is improved by about 1 cm. Moreover, assessed by shipborne gravity and SIO V32.1, the RMS of gravity anomalies from SSH at 1 Hz assessed by SIO V32.1 is slightly better than that of SSHs at varying length scales. Thus, SSHs of ICESat-2 resampled at 1 Hz are used to recover global marine gravity anomalies.

Table 3 The quality of ICESat-2 SSHs and gravity models recovered from SSHs at varying length scales and at resampled 1 Hz

| SSHs at different sampling frequency | The number of SSHs | The RMS of SSH crossover discrepancies after adjustment (m) | The difference between Gravity anomalies recovered from ICESat-2 and Shipborne gravity (mGal) | | The difference between Gravity anomalies recovered from ICESat-2 and SIO V32.1 (mGal) | |
|---|---|---|---|---|---|---|
| | | | \|Max\| | RMS | \|Max\| | RMS |
| SSHs at varying length scales | 1 457 596 | 0.124 | 50.02 | 5.44 | 52.30 | 3.06 |
| SSHs at 1 Hz | 854 533 | 0.115 | 49.54 | 5.42 | 52.01 | 2.89 |

2. The geoid height is derived from SSH observations, with the dynamic topography removed as the non-geoidal signal. The authors use the MDT_CNES_CLS18 model with a grid resolution of 7.5 arcmin to remove this signal. Given that the average along-track ground distance of altimeter data is about 7 km (1 arcmin), it is crucial to understand how the removal value from MDT_CNES_CLS18 is determined. Detailed methodology on this aspect would enhance comprehension.

Reply: Thank you for your thoughtful comments. The dynamic topography is an essential non-geodetic signal. It should be removed from SSH observations to obtain the geoid height. In general, the mean dynamic topography (MDT) model is used in order to reduce the effect of dynamic topography. The MDT_CNES_CLS18 model (or other model) is a regular grid data of 7.5'×7.5'. The SSH observations is at 1 Hz sampled frequency. The removed value of MDT at corresponding position of SSHs is derived by the bivariate spline interpolation.

The sentence is rephrased:

Secondly, the residual geoid heights are determined by removing the mean dynamic topography model and the reference geoid model from corrected SSHs. The removed valve of MDT_CNES_CLS18 (Mulet et al. 2021) or geoid model at corresponding position of SSHs is derived by the bivariate spline interpolation.

3. Filtering is crucial for the fusion of multi-altimeter data. The filter radius for along-track radar and laser altimeter data is noted as 7 km. However, details regarding the filter radius and its application are missing. Is the filtering applied in the along-track direction or in the spatial domain? Is the purpose to reduce spatial high-frequency error or to mitigate the temporal SSH signal? Clarification on these points is essential for a thorough understanding and accurate interpretation of the results.

Reply: Thanks for your comments. Gaussian filter is applied in along-track SSHs from radar and laser altimeter data in order to reduce the influence of sea surface temporal variability and high-frequency noise. The response function is selected as

$$r(d) = \exp(-\frac{d^2}{2s^2})$$

where $d$ is the spherical distance between two data points. $s$ is the radius of the convolution window

and is defined as the filtering parameter. Filtered values are obtained by convolving all SSHs in the window of radius $s$ with the response function.

In section 4.1, we have discussed the filter radius for the gravity anomalies recovered from ICESat-2. According to the average along-track ground distance, the filtering radius with a multiple of 7 km is used to recover gravity anomaly model. To conclude, the filtering radius of 7 km is selected for the gravity anomalies recovery from ICESat-2 along-track SSHs.

For radar altimeter data, the filter radius is generally consistent with the recovery of SDUST2021GRA gravity anomaly model (Zhu et al., 2022). Thus, it is not presents in this manuscript.

Zhu, C., Guo, J., Yuan, J., Li, Z., Liu, X., and Gao, J.: SDUST2021GRA: global marine gravity anomaly model recovered from Ka-band and Ku-band satellite altimeter data, Earth Syst. Sci. Data, 14, 4589–4606, https://doi.org/10.5194/essd-14-4589-2022, 2022.

4. Section 3.2 outlines the steps for determining cross-track geoid gradients. However, merely listing the processing steps can lead to ambiguity. For instance, the phrase "one track with good observation is selected as the reference altimeter data" is unclear. Does "good" refer to accuracy or the number of observations? Providing the corresponding formula or a detailed processing flowchart would greatly improve clarity.

Thanks very much for your valuable comments. We have revised the processing steps for determining cross-track geoid gradients (GGs), and added the corresponding formula to determine the cross-track GGs from any two of three beams observations. The sentence in Section 3.2 is rephrased:

Because three beams of ICESat-2 observations are not exactly simultaneous, the cross-track GG is determined, according to the following steps. (1) One beam with good observations (maximum number) from two beams altimeter data is selected as the reference altimeter data. (2) Based on the reference beam observations, the cross-track GG is determined within an azimuth threshold. (3) If the number of GGs is more than one on each reference observation, only the cross-track GG with an azimuth closet to perpendicular to the orbit inclination is used to recover gravity anomalies. A schematic diagram of determining the cross-track gt13 GGs from ICESat-2 altimeter data is shown in Fig. 3. The cross-track GG determination strategy is defined as follows:

$$
\begin{cases}
\text{Reference\_beam} = \text{Max}[\text{Num}_{\text{gt1}}, \text{Num}_{\text{gt3}}] \\
\left| T_i - T_{ref} \right| \leq T\_Threshold \\
\left| \alpha_{GG,i} - \alpha_{ref\_inc} \right| \leq A\_Threshold \\
\text{Cross\_track\_GG} = \text{Min}[\alpha_{GG,i} - \alpha_{ref\_inc}]
\end{cases}
\tag{7}
$$

where $\text{Num}_{\text{gt1}}$, $\text{Num}_{\text{gt2}}$, and $\text{Num}_{\text{gt3}}$ are the number of each beam observations, respectively. $T_{ref}$ is the observation time of reference beam, $T_i$ is the observation time of the other beam, $\alpha_{GG,i}$ is the azimuth of GG derived from two-beam observations at the number $i$. $\alpha_{ref\_inc}$ is a reference azimuth perpendicular to the orbit inclination. $T\_Threshold$ is a time threshold, and 1 s is selected as time

threshold to reduce the effect of random errors, $A\_Threshold$ is an azimuth threshold, $\pi/4$ serves as a azimuth threshold to obtain GGs with azimuth toward east-west direction.

[Figure]

Figure 3 The schematic diagram of determining the cross-track geoid gradients from gt1 and gt3 beams of ICESat-2

5. The precision of the gravity anomaly model is assessed by comparing it to shipborne gravity anomalies. While the RMS is 4-5 mGal in global oceans and low-middle latitudes, it approaches 10 mGal in high-latitude regions. An explanation for the lower precision in high-latitude regions is necessary to understand this discrepancy.

Reply: Thanks for your comments. For this large difference, I think there are two main reasons. First, the number of shipborne gravity data is small, as shown in global shipborne gravity distribution (Fig. 2). The precision of shipborne gravity in high latitude region is probably lower than that of in low-middle latitudes. In some local high latitude regions, the RMS of difference is large than 10 mGal, as shown in the assessment of SDUST2021GRA (Zhu et al., 2022). Second, the reduced quantity and low precision of SSHs also degraded the accuracy of the recovered gravity anomaly model. The average number of GGs from all altimeter data within the 1′×1′ grid is calculated, as presented in Fig. 8. In addition, we also compared SDUST2022GRA and shipborne gravity in local high latitude region (140°W-170°W, 80°S-66°S). The statistic of difference is shown in Table 11. The RMS is about 5.14 mGal, which consistent with the assessment in low-middle latitudes.

[Figure]

Figure 8 The number of SSHs within the 1′×1′ grid in different local regions. a: open ocean with average number of 3.5. b: the high latitude region with average number of 2.1. c: coastal region with average number of 1.9.

Table 11 The Mean and RMS of difference between gravity anomaly models and shipborne gravity in local regions (Unit: mGal)

| Local region | | NSOAS22 | | DTU17 | | SIO V32.1 | | SDUST2022GRA | |
|---|---|---|---|---|---|---|---|---|---|
| | | Mean | RMS | Mean | RMS | Mean | RMS | Mean | RMS |
| Region A1 | Open | 0.15 | 3.58 | 0.10 | 3.24 | -0.10 | 3.15 | 0.20 | 3.04 |
| Region A2 | ocean | -0.41 | 5.13 | -0.41 | 4.29 | 0.14 | 3.78 | 0.01 | 4.01 |
| Region B1 | | -1.51 | 8.47 | -1.81 | 7.21 | 0.10 | 6.25 | -0.16 | 6.08 |
| Region B2 | Coastal | -0.86 | 10.66 | -1.41 | 10.33 | -0.56 | 7.85 | -0.57 | 7.69 |
| Region B3 | region | 0.10 | 12.12 | -1.24 | 11.25 | -0.67 | 10.32 | -0.68 | 10.10 |
| Region C1 | High-latitude region | 0.33 | 5.86 | 0.15 | 5.36 | 0.12 | 5.38 | 0.12 | 5.14 |

6. Spatial resolution is a crucial index of the gravity anomaly model. Cross-spectral analysis is typically used to determine the wavelength of the model by comparing it to shipborne gravity. The paper derives three results from shipborne gravity, yet the wavelengths from the gravity anomaly model differ. An explanation for this variance and what determines the spatial resolution of the model would be beneficial.

Reply: Thank you for your thoughtful comments. The spatial resolution is certainly a crucial index of the altimeter-recovered gravity anomaly model. The wavelength derived from SDUST2022GRA with a CMS value of 0.5 is 18.6 km, 20.7 km, and 20.4 in a local open ocean region, high latitude region, and coastal region, respectively. There are two main reasons for the difference. First, the assessment is related to the data interval of shipborne gravity. Three cruises shipborne gravity anomalies are used to determine the spatial resolution of SDUST2022GRA, SIO V32.1, and DTU17, as shown in Fig. 5. In addition, the spatial resolution of gravity anomaly model is mainly determined by altimeter data resolution and density. In different regions, there are also slight variations in the density of altimeter data, as shown in Fig.8.

[Figure]

Figure 5 Shipborne gravity (used to determine CMS) of different cruises. a: the jare33l1 with average distance interval of 0.45 km. b:. the ew9201 with average distance interval of 0.80 km. c: the moce05mv with average distance interval of 0.22 km.

[Figure]

Figure 8 The number of SSHs within the 1′×1′ grid in different local regions. a: open ocean with average number of 3.5. b: the high latitude region with average number of 2.1. c: coastal region with average number of 1.9.

Minor edits:

7.  I recommend that the authors improve the clarity and readability of the manuscript by refining the English language usage
    Line 17: "across-track direction" should be "cross-track direction".
    Line 115: SARAL/Altika is operate in Ka-band not Ku-band, please correct it.

Reply: Thanks for your thoughtful comments. This 'across-track' is corrected to 'cross-track' as the suggestion in all sections. And we corrected the sentence 'SARAL/Altika is operate in Ka-band not Ku-band'.

8.  Line 146: "a quadratic polynomial was used to correct long wavelength system error". This statement is not accurate. The quadratic polynomial is used for shipborne gravity from each cruise in order to correct system bias relative to the gravity reference field.

Reply: Thank you very much for your suggestion. The sentence is rephrased:

Then, for gravity anomalies from each cruise, the system bias caused by the drift of the gravimeter was corrected by a quadratic polynomial, which is detail in Hwang and Parsons (1995).

9.  Line 252: "the maximum distance of along-track …", what is the maximum distance?

Reply: Thanks for your suggestion. The sentence is rephrased:

For SSHs of ICESat-2, the average ground distance of along-track adjacent observations is about 7 km, so the filtering radius with a multiple of 7 km is applied to recover marine gravity anomalies from along-track altimeter data.

10. Line 265: In Table 5, please specify the unit of geoid gradients?

Reply: Thanks for your comment. We scrutinized the unit of geoid gradients. The unit in Table 5 is corrected.

Table 5 The number and STD of residual GGs from ICESat-2

| Residual GGs | gt1 | gt2 | gt3 | gt12 | gt23 | gt13 |
|---|---|---|---|---|---|---|
| Number | 302407 | 250988 | 301138 | 202492 | 200312 | 209769 |
| STD(urad) | 1.93 | 1.88 | 1.91 | 2.66 | 2.75 | 1.94 |

11. Line 369: The percentage contribution formula is not explained clearly about the use of the variable

RMS. Please add an description.

Reply: Thank you for your valuable comments. The percentage contribution is redefined as $\frac{RMS_{\text{SDUST2022GRA}} - RMS_{\text{SDUST2021GRA}}}{RMS_{\text{SDUST2022GRA}}} \times 100\%$ and the improvement is recalculated. The sentence is rephrased: the percentage contribution of ICESat-2 to the improvement of gravity anomaly model is 4.3% in low-middle latitude regions, and it is increasing in coastal regions.

12. Line 385: In Table 14, the regions A, B, C, D, E, F are not mentioned in the text. please specify the used region.

Reply: Thanks for your comments. We corrected the description of region in the Table 14.

Table 14 The percentage contribution of ICESat-2 altimeter data in different local regions

| Local region | RMS_SDUST2021GRA (mGal) | RMS_SDUST2022GRA (mGal) | △RMS (mGal) | Percentage Contribution |
|---|---|---|---|---|
| Region A1 | 3.12 | 3.04 | 0.08 | 2.5% |
| Region A2 | 4.07 | 4.01 | 0.06 | 1.5% |
| Region B1 | 6.40 | 6.08 | 0.32 | 5.0% |
| Region B2 | 7.98 | 7.69 | 0.28 | 3.5% |
| Region B3 | 10.51 | 10.10 | 0.41 | 3.9% |
| Region C1 | 5.32 | 5.14 | 0.18 | 3.3% |

---

## Author Comment (AC2)

**Comment:** This paper derived the global marine gravity model, SDUST2022GRA, by combining altimeter data from nadir-looking satellites and from ICESat-2. The cross-track geoid gradients were determined using the three-pair laser beams of ICESat-2, a capability not available from nadir-looking altimeters. These cross-track geoid gradients, which are primarily oriented in the east-west direction, are instrumental in improving the accuracy of the east-wast components of geoid gradients, thereby enhancing the marine gravity model. This paper is interesting and worth publishing after addressing the key points identified herein.

Reply: Thank you for your comments and meticulous check of our manuscript that make the manuscript more interesting and informative. Below, please find our point to point responses to the comments in the list of revisions.

**Comment 1** Lines 210-215: When determining the cross-track geoid gradients, the authors only used the data with 'the closest time'. Although time-related signals affect ICESat-2 SSHs, the time for the SSHs from the ICESat-2's three-pair beams is close. Ignoring the time factor may yield more results.

Reply: Thank you very much for your comments. The method of determining the cross-track geoid gradients is rephrased:

Because three beams of ICESat-2 observations are not exactly simultaneous, the cross-track GG is determined, according to the following steps. (1) One track with good observations (maximum number of each beam) from two-beam altimeter data is selected as the reference altimeter data. (2) Based on the reference beam observations, the cross-track GG is determined within a time and azimuth threshold. (3) If the number of GGs is more than one on each reference observation, only the cross-track GG with an azimuth closet to perpendicular to the orbit inclination is used to recover gravity anomalies. A schematic diagram of determining the cross-track gt13 GGs from ICESat-2 altimeter data is shown in Fig. 3. The cross-track GG determination strategy is defined as follows:

$$\begin{cases} \text{Reference\_beam} = \text{Max}[\text{Num}_{gt1}, \text{Num}_{gt3}] \\ \left| T_i - T_{ref} \right| \leq T\_Threshold \\ \left| \alpha_{GG,i} - \alpha_{ref\_inc} \right| \leq A\_Threshold \\ \text{Cross\_track\_GG} = \text{Min}[\alpha_{GG,i} - \alpha_{ref\_inc}] \end{cases} \quad (7)$$

where $\text{Num}_{gt1}, \text{Num}_{gt2}$, and $\text{Num}_{gt3}$ are the number of each beam observations, respectively. $T_{ref}$ is the observation time of reference beam, $T_i$ is the observation time of the other beam, $\alpha_{GG,i}$ is the azimuth of GG derived from two-beam observations at the number $i$. $\alpha_{ref\_inc}$ is a reference azimuth perpendicular to the orbit inclination. $T\_Threshold$ is a time threshold, and 1 s is selected as time threshold to reduce the effect of random errors, $A\_Threshold$ is an azimuth threshold, $\pi/4$ serves as a azimuth threshold to obtain GGs with azimuth toward east-west direction.

[Figure]

Figure 3 The schematic diagram of determining the cross-track geoid gradients from gt1 and gt3 beams of ICESat-2

For the determination of cross-track GG of ICESat-2, it is necessary to select the associated SSHs from different beam observations. Otherwise, a cross-track GG with an azimuth that inclined to the north direction may not be able to mitigate the unbalanced accuracy of DOV.

**Comment 2** Figure 3 and Table 5: Three types of cross-track geoid gradients (g12, g23, and g13) were obtained (Fig. 3). The accuracy of g13 was much higher than that of g12 and g23 (Table 5). Therefore, only g13 was used to derive the marine gravity model. However, the differences in the STD for the three types of geoid gradients are due to the distance rather than the accuracy of ICESat-2 SSHs. A more reasonable explanation should be provided.

Reply: Thank you very much for your valuable suggestions. We used the along-track GGs and gt13 cross-track GGs to recovering marine gravity anomaly model. The main reason for exclusively using the gt13 cross-track GGs is that the configuration yields a high accuracy for gravity anomalies recovery compared to other combinations. Table 4 provides statistic on the difference between gravity anomalies recovered from ICESat-2 and shipborne gravity. Our analysis indicates that combining gt13 cross-track GGs results in better accuracy than combining gt12 or gt23 cross-track GGs.

For this reason, we analyses the number of observations from three beams, the precision of SSHs and GGs. The precision of SSHs is a critical factor influencing the precision of GGs. The RMS of SSH crossover discrepancies from three beam observations is listed in Table 6. While the precision of SSHs from the gt2 beam observation is slightly superior to that from the gt1 or gt3, it is not straightforward to determine that the precision of cross-track GGs. The precision of GGs is not only related to the precision of SSHs, but also to the distance between the two points. Furthermore, we analyzed the quality (number and standard deviation) of along-track and cross-track GGs, as shown in Table 5. The STD of difference between gt13 GGs and the reference gravity field is closer to that of along-track GGs than gt12 and gt23. Additionally, the number of gt2 beam observation is less than gt1 or gt3 beam observations, resulting in the number of gt13 cross-track GGs being more than that other cases. Therefore, the combination of along-track and gt13 cross-track GGs is used to recover marine gravity anomalies.

Table 4 Differences between ICESat-2 altimeter-derived gravity and ship-borne gravity (Unit: mGal)

| Gravity anomaly model | Max | Min | Mean | STD | RMS |
|---|---|---|---|---|---|

| | | | | | |
|---|---|---|---|---|---|
| gt1+gt2+gt3 | 50.83 | -48.28 | -0.13 | 5.56 | 5.56 |
| gt12+gt1+gt2+gt3 | 49.35 | -48.18 | -0.10 | 5.66 | 5.66 |
| gt23+gt1+gt2+gt3 | 54.92 | -54.98 | -0.06 | 5.70 | 5.70 |
| gt12+gt23+gt1+gt2+gt3 | 47.07 | -46.75 | -0.07 | 5.65 | 5.65 |
| gt13+gt1+gt2+gt3 | 49.54 | -48.05 | -0.03 | 5.42 | 5.42 |

Table 5 The number and STD of residual GGs from ICESat-2

| Residual GGs | gt1 | gt2 | gt3 | gt12 | gt23 | gt13 |
|---|---|---|---|---|---|---|
| Number | 302407 | 250988 | 301138 | 202492 | 200312 | 209769 |
| STD(urad) | 1.93 | 1.88 | 1.91 | 2.66 | 2.75 | 1.94 |

Table 6 The RMS of SSH crossover discrepancies

| Altimetry | Satellite Mission | Average along-track ground distance (km) | Crossover discrepancies (30 d) | |
|---|---|---|---|---|
| | | | RMS before adjustment (m) | RMS after adjustment (m) |
| Laser altimetry | ICESat-2/gt1 | 7.1 | 0.131 | 0.117 |
| | ICESat-2/gt2 | 7.1 | 0.128 | 0.109 |
| | ICESat-2/gt3 | 7.1 | 0.138 | 0.119 |
| GM (Radar altimetry) | SARAL/DP | 7.0 | 0.110 | 0.085 |
| | Cryosat-2 | 6.4 | 0.082 | 0.060 |
| | H2A | 6.5 | 0.103 | 0.076 |
| | J2 | 5.8 | 0.114 | 0.088 |
| | J1 | 5.8 | 0.108 | 0.079 |
| | E1 | 6.4 | 0.117 | 0.097 |
| ERM (Radar altimetry) | Sentinel-6A SAR | 5.8 | 0.022 | 0.013 |
| | Sentinel-3A SAR | 6.7 | 0.027 | 0.018 |
| | Sentinel-3B SAR | 6.7 | 0.035 | 0.026 |
| | SARAL | 7.0 | 0.034 | 0.020 |
| | HY-2A | 6.5 | 0.030 | 0.020 |
| | HY-2B | 6.5 | 0.032 | 0.024 |
| | T/P-Jason_A | 5.9 | 0.027 | 0.018 |
| | T/P-Jason_B | 5.9 | 0.026 | 0.019 |
| | Envisat_A | 7.5 | 0.033 | 0.022 |
| | Envisat_B | 7.5 | 0.042 | 0.024 |
| | ERS-2 | 6.6 | 0.040 | 0.034 |
| | GFO | 6.7 | 0.034 | 0.019 |

**Specific and minor comments**

1. Line 15: In this manuscript, 'across-track' should be corrected to 'cross-track'.

Reply: Thanks. This 'across-track' is corrected to 'cross-track' as the suggestion in all sections.

2. Line 30: The 13% improvement in the marine gravity field is significantly impressive, but ICESat-2 only improved by 0.12 mgal, which could mislead readers.

Reply: Thank you for your valuable comments. The percentage contribution is redefined as $\dfrac{RMS_{\text{SDUST2022GRA}} - RMS_{\text{SDUST2021GRA}}}{RMS_{\text{SDUST2022GRA}}} \times 100\%$ and the improvement is recalculated. The sentence is rephrased: the percentage contribution of ICESat-2 to the improvement of gravity anomaly model is 4.3% in low-middle latitude regions, and it is increasing in coastal regions.

3. Line 40: The authors confused the 'DOV' and 'geoid gradient (GG)' concepts. In this manuscript, DOV is geoid gradients, but they are opposite. The along-track GGs (and cross-track GG from ICESat-2) are used to determine the north and east components of GG on a regular grid using LSC. Therefore, this sentence should be:'Normally, the north and east components of deflection of the vertical (DOV) on a regular grid, derived from along-track geoid gradients, is ...'

Reply: Thanks for your thoughtful comments. The sentence is rephrased:

Normally, the north-south component and east-west component of deflection of the vertical (DOV) on a regular grid, derived from along-track geoid gradients (GGs), is used to recover marine gravity anomaly model by inverse Vening-Meinesz formula or Laplace's equation (Sandwell and Smith 1997; Hwang et al., 2002).

Because of the satellite ground-tracks inclination of the north-south direction, the precision of the north component of the altimeter-derived DOV model is generally higher than the east component (Che et al., 2021; Jin et al. 2022). The unbalanced accuracy of DOV components severely restricts the improvement of the gravity anomaly model (Hwang 1998, Annan and Wan 2021).

4. Line 140: Should it be 'satellite-derived gravity anomaly models'?

Reply: Thank you for your suggestion. The sentence is rephrased:

In general, shipborne gravity anomalies have a higher accuracy and spatial resolution than the altimeter-derived gravity anomaly model on ship routes.

5. Line 143: Please modify this sentence to: The gross errors in the shipborne gravity data were removed. I would like to know if the shipborne data, which removed the long-wavelength errors based on XGM2019e, were used to assess all the marine gravity models described in this manuscript.

Reply: Thank you very much for your good comments. The sentence is rephrased:

The gross errors in the shipborne gravity data were removed. The shipborne gravity data, which removed outliers and long-wavelength errors based on XGM2019e, were used to assess all marine gravity anomaly models. Table 10 presents the difference between altimeter-derived gravity anomaly models (NSOAS22, DTU17 SIO V32.1, SDUST2022GRA) and all shipborne gravity anomalies. Table 13 presents the RMS of difference between SDUST2021GRA and all shipborne gravity anomalies.

Table 10 The difference between gravity anomaly models and global shipborne gravity (Unit: mGal)

| Region | Model | Max | Min | Mean | STD | RMS |
|---|---|---|---|---|---|---|
| Global ocean [80°S, 82°N] | NSOAS22 | 99.46 | -81.17 | -0.10 | 5.73 | 5.73 |
| | DTU17 | 99.25 | -71.85 | -0.13 | 5.42 | 5.42 |
| | SIO V32.1 | 77.17 | -86.24 | -0.10 | 5.18 | 5.18 |
| | SDUST2022GRA | 96.79 | -68.51 | -0.08 | 5.07 | 5.07 |
| Low-middle latitude | NSOAS22 | 78.04 | -81.17 | -0.07 | 5.26 | 5.26 |

| | | | | | | |
|---|---|---|---|---|---|---|
| regions | DTU17 | 78.44 | -71.85 | -0.12 | 4.89 | 4.89 |
| [60°S, 60°N] | SIO V32.1 | 76.25 | -86.23 | -0.06 | 4.65 | 4.65 |
| | SDUST2022GRA | 64.44 | -67.00 | -0.09 | 4.43 | 4.43 |
| High-latitude regions [80°S, 60°S)& (60°N, 82°N ] | NSOAS22 | 99.46 | -70.56 | -0.47 | 9.76 | 9.77 |
| | DTU17 | 99.25 | -68.48 | -0.25 | 9.82 | 9.82 |
| | SIO V32.1 | 77.17 | -76.54 | -0.51 | 9.53 | 9.54 |
| | SDUST2022GRA | 96.79 | -68.48 | -0.26 | 9.69 | 9.69 |

Table 13 The percentage contribution of ICESat-2 altimeter data in global ocean region

| Region | RMS$_{SDUST2021GRA}$ (mGal) | RMS$_{SDUST2022GRA}$ (mGal) | $\triangle$RMS (mGal) | Percentage Contribution |
|---|---|---|---|---|
| Global ocean | 5.19 | 5.07 | 0.12 | 2.3% |
| Low-middle latitude regions | 4.63 | 4.43 | 0.20 | 4.3% |
| High-latitude regions | 9.73 | 9.69 | 0.04 | 0.4% |

6. Line 150. Why is the accuracy of shipborne data 2.82 mGal?

Reply: Thanks for your comments. The precision of shipborne gravity data is not provided along with the gravity data. Thus, the precision of shipborne gravity is verified by the discrepancies of gravity anomalies at crossover points. The RMS of discrepancies is 3.99 mGal. The precision of shipborne gravity anomalies, about 2.82 mGal, is derived by dividing RMS by the square root of two based on the error propagation law. It is generally consistent with the shipborne gravimeter measurements of 1-3 mGal magnitude (Zaki et al. 2022).

7. Please provide this abbreviation 'geoid gradient'. when it first appear.
   Please provide the full name of STD. Cnn is not described in this manuscript. Please change it to 'error variance of GGs'.

Reply: Thanks for your suggestion. We add the abbreviation 'geoid gradient', when it first appear in Line 39. The sentence is rephrased: The residual along-track GG is derived by

We apologize for the missing of full name of STD and the description of Cnn. The sentence is rephrased: where is the standard deviation (STD) of GGs to determine the error variance (Cnn in LSC) of GGs.

8. In equation (3), The two parameters (t0 and t1) are very important. Please provide specific values.

$$f(t) = a_0 + a_1(t - t_0) + \sum_{i=1}^{n} \left[ C_i \cos(i\omega(t - t_0)) + S_i \sin(i\omega(t - t_0)) \right] \qquad (3)$$

Reply: Thank you very much for your comments. t0 and t1 are two parameter used to determined the angular frequency in the trigonometric polynomial. t0 and t1 are the beginning and end observation times of each ground track, respectively. For each track observations, the value is not the same. The detail of crossover adjustment is described in Huang et al. (2008).

9. How to compute this parameter? The initial value $D_{\Delta g,0}$.

The termination condition of the iteration is that the difference between the adjacent error of GG is less than a threshold. Please provide specific.

Reply: Thanks for your thoughtful comments. The initial value $D_{\Delta g,0}$ is determined using

The each altimeter-derived gravity anomaly model is recovered from the initial error of GGs derived by the RMS of crossover discrepancies, include the Ku-band and Ka-band altimeter data. The precision of altimeter-derived gravity anomalies can be calculated by

$$D_{model\_ship} + v = D_{\Delta g} + D_{shipborne}$$

Where $D_{model\_ship}$ represent the variance of difference between altimeter-derived gravity anomaly model and shipborne gravity anomalies, $D_{shipborne}$ represent the variance of shipborne gravity anomalies, $D_{\Delta g}$ is the error variance of altimeter-derived gravity. The SARAL(Ka-band)-derived gravity anomaly model is recovered from the initial error of GGs derived by the RMS of crossover discrepancies. Then, the initial value $D_{\Delta g,0}$ is determined by above equation.

The termination condition of the iteration is less than a threshold. The threshold is determined from the RMS of fitted residuals of parameters solution ($\beta_0$ and $\beta_1$). The Threshold is 0.04 mGal, which provided after solving ($\beta_0$ and $\beta_1$) in Section 4.2.

10. Do you want to present that 'We presented a method for processing multi-beam observations from ICESat-2' ?

Reply: Thank you very much for your comments. The sentence is rephrased:

The cross-track GGs processing method is presented from ICESat-2 multiple beam observations.

11. The used LSC for the determination of DOV is.  Please add the reference: (Hwang and Parsons, 1995)

    Lines 230 and 235, please add Reference.

    Equations (9) and (11) should not be provided as an independent formula.

Reply: Thank you very much for your valuable suggestion. We corrected the sentence and added Reference. In addition, Equations (9) and (11) are provided in the text not as an independent formula.

The sentence is rephrased:

We determined the DOV components by LSC as (Hwang and Parsons, 1995)

In Lines 230 and 235, we added References. The sentence is rephrased:

Therefore, the covariance matrices ($C_{\xi_e}$, $C_{\eta e}$ and $C_{ee}$) are obtained by (Hwang and Parsons, 1995)

The gravity anomaly model is recovered by the inverse Vening-Meinesz formula as (Hwang, 1998)

Reference

Hwang, C.: Inverse Vening Meinesz formula and deflection-geoid formula: applications to the predictions of gravity and geoid over the South China Sea, J. Geodesy, 72(5), 304-312, https://doi.org/10.1007/s001900050169, 1998.

Hwang, C., and Parsons, B.: Gravity anomalies derived from Seasat, Geosat, ERS-1 and

TOPEX/POSEIDON altimetry and ship gravity: a case study over the Reykjanes Ridge, Geophys. J. Int., 122(2), 551-568, https://doi.org/10.1111/j.1365-246X.1995.tb07013.x, 1995.

12. When resampling the ICESat-2 data, would additional errors caused by resampling or interpolation appear?

Reply: Thanks for your suggestion. The resampling is an important method for altimeter data processing. We added the introduction about resampled method in Section 3.2.

The ICESat-2 SSH observations at varying length scales is resampled at 1 Hz for each beam in order to obtain a uniform distribution of SSHs. In the resampling, we fitted SSHs at varying length scale by a quadratic polynomial in latitude to reduce the effect of the high-frequency noise and outlier. Each of the 1-s SSHs is used to solve polynomial coefficients and then produced SSHs in the median of the latitude. When the number of observations is less than the minimum number for the solution of polynomial coefficients, the 1-s SSHs are averaged directly to 1 Hz. The used quadratic polynomial function of latitude is (Yu and Hwang, 202)

$$l_i + v_i = a\varphi_i^2 + b\varphi_i + c \qquad (6)$$

where $l_i$ is the SSH observation at point $i$ with in a time threshold, $v_i$ is the residual at point $i$, $\varphi_i$ is the latitude at point $i$, and $a$, $b$, $c$ are the coefficients of the quadratic polynomial.

13. In table 4. What does 'STD'mean? What is the difference between 'STD' and 'RMS'?

Reply: Thank you very much for your comments. The STD accounts for the deviation of individual data points from the mean, where as RMS accounts for the absolute magnitude of those data points. In there, the STD and RMS is determined from the difference between ICESat-2 recovered gravity anomalies and shipborne gravity anomalies. The Table 4 is corrected as:

Table 4 Differences between ICESat-2 altimeter-derived gravity and ship-borne gravity (Unit: mGal)

| Gravity anomaly model | Max | Min | Mean | STD | RMS |
|---|---|---|---|---|---|
| gt1+gt2+gt3 | 50.83 | -48.28 | -0.13 | 5.56 | 5.56 |
| gt12+gt1+gt2+gt3 | 49.35 | -48.18 | -0.10 | 5.66 | 5.66 |
| gt23+gt1+gt2+gt3 | 54.92 | -54.98 | -0.06 | 5.70 | 5.70 |
| gt12+gt23+gt1+gt2+gt3 | 47.07 | -46.75 | -0.07 | 5.65 | 5.65 |
| gt13+gt1+gt2+gt3 | 49.54 | -48.05 | -0.03 | 5.42 | 5.42 |

14. The STD of g13 is much smaller than those of g12 and g23. The reason is that the distance of g13 (please see Fig. 3 and Eq. (1)) is half of that of g12 and g23. But we could not conclude that g13 is much accurate than g12 and g23.

Reply: Thanks for your thoughtful comments. The error variance of GG from each altimeter data can be derived using the error propagation law of Eq. (1) while ignoring the distance error of two points, as

$$m_e^2 = \frac{m_{ssh,pt1}^2 + m_{ssh,pt2}^2}{d_{pt1\_pt2}^2} \qquad (2)$$

Thus, the GGs error is correlated with the distance and SSH observations error. The STD of g13 is much smaller than those of g12 and g23, but it could not conclude that gt13 is much accurate than gt12 and gt23. We corrected the explanation. The sentence is rephrased:

The total amount of gt13 GGs is generally consistent with gt12 and gt23, but the STD of difference between gt13 GGs and the reference gravity field is close to that of along-track GGs than gt12 and gt23.

15. The contribution of ICESat-2 to the marine gravity model is the difference between the RMSs of SDUST2021GRA and SDUST2022GRA. It has nothing to do with the reference field. The 13% improvement in the marine gravity field is significantly impressive, but ICESat-2 only improved by 0.12 mgal, which could mislead readers.

Reply: Thank you very much for your valuable comments. The percentage contribution is redefined. The sentence is rephrased:

The percentage contribution of ICESat-2 to the improvement of the gravity anomaly model is defined as

$$\frac{RMS_{\text{SDUST2022GRA}} - RMS_{\text{SDUST2021GRA}}}{RMS_{\text{SDUST2022GRA}}} \times 100\%$$ representing the ratio of the improvement of the

gravity model recovered by incorporating ICESat-2 to the improvement of the gravity model recovered from all altimeter data, as shown in Table 13.

Table 13 The percentage contribution of ICESat-2 altimeter data in global ocean region

| Region | RMS$_{\text{SDUST2021GRA}}$ (mGal) | RMS$_{\text{SDUST2022GRA}}$ (mGal) | ΔRMS (mGal) | Percentage Contribution |
|---|---|---|---|---|
| Global ocean | 5.19 | 5.07 | 0.12 | 2.3% |
| Low-middle latitude regions | 4.63 | 4.43 | 0.20 | 4.3% |
| High-latitude regions | 9.73 | 9.69 | 0.04 | 0.4% |